# O-MAPL: Offline Multi-agent Preference Learning

**The Viet Bui** [1]  **Tien Mai** [1]  **Hong Thanh Nguyen** [2]

## Abstract

Inferring reward functions from demonstrations is a key challenge in reinforcement learning (RL), particularly in multi-agent RL (MARL), where large joint state-action spaces and complex inter-agent interactions complicate the task. While prior single-agent studies have explored recovering reward functions and policies from human preferences, similar work in MARL is limited. Existing methods often involve separate stages of supervised reward learning and MARL algorithms, leading to unstable training. In this work, we introduce a novel end-to-end preference-based learning framework for cooperative MARL, leveraging the underlying connection between reward functions and soft Q-functions. Our approach uses a carefully-designed multi-agent value decomposition strategy to improve training efficiency. Extensive experiments on SMAC and MAMuJoCo benchmarks show that our algorithm outperforms existing methods across various tasks.

## 1. Introduction

Reinforcement learning (RL) has been instrumental in a wide range of decision making tasks, where agents gradually learn to operate effectively through interactions with their environment (Levine et al., 2016; Silver et al., 2017; Kalashnikov et al., 2018; Haydari & Yılmaz, 2020). Typically, when an agent takes an action, it receives feedback in the form of reward signals, enabling it to adjust or revise its action plan (i.e., policy). However, designing an appropriate reward function is a significant challenge in many real-world domains. While essential for training successful RL agents, reward design often requires extensive instrumentation or engineering (Yahya et al., 2017; Schenck & Fox, 2017; Peng et al., 2020; Yu et al., 2020; Zhu et al., 2020). Moreover, such reward functions can be exploited

---
*Equal contribution [1]Singapore Management University, Singapore [2]University of Oregon Eugene, Oregon, United States. Correspondence to: Tien Mai <atmai@smu.edu.sg>.

*Proceedings of the $42^{nd}$ International Conference on Machine Learning*, Vancouver, Canada. PMLR 267, 2025. Copyright 2025 by the author(s).

by RL algorithms, which might find ways to achieve high expected returns by inducing unexpected or undesirable behaviors (Hadfield-Menell et al., 2017; Turner et al., 2020).

To address these challenges, many RL studies have relaxed the reward structure by using sparse rewards, where agents receive feedback periodically (Arjona-Medina et al., 2019; Ren et al., 2021; Zhang et al., 2024b). While this reduces the need for dense reward signals, it is often insufficient to train effective agents in complex domains. Alternatively, imitation learning (IL) has been explored, where agents learn to mimic an expert's policy from demonstrations, without explicit reward signals (Ho & Ermon, 2016; Fu et al., 2017; Garg et al., 2021; Mai et al., 2024). However, achieving expert-level performance with IL requires a large amount of expert data, which can be costly and difficult to obtain.

A recent promising approach is to train agents using human preference data, a more resource-efficient form of feedback called Reinforcement Learning from Human Feedback (RLHF). This allows agents to learn behaviors aligned with human intentions. RLHF has proven effective in both single-agent control (Christiano et al., 2017; Mukherjee et al., 2024; Lee et al., 2021; Shin et al., 2023; Hejna & Sadigh, 2024) and natural language tasks (Stiennon et al., 2020; Ouyang et al., 2022; Rafailov et al., 2024). However, RLHF in multi-agent environments is still underexplored, as simply extending single-agent methods is insufficient due to the complex interdependencies between agents' policies.

Only a few recent studies have developed preference-based RL algorithms for multi-agent settings (Kang et al., 2024; Zhang et al., 2024a), typically using a two-phase learning framework: first, preference data trains a reward model, and then the policy is optimized. However, this approach has two main drawbacks: (i) it requires large preference datasets to cover the state and action spaces, and (ii) misalignment between the two phases can degrade policy quality.

In this work, we investigate multi-agent preference-based RL (PbRL), focusing on the offline learning setting where agents do not interact with the environment but instead have access to an offline dataset of pairwise trajectory preferences. Unlike previous studies in multi-agent PbRL, we propose an end-to-end learning approach that directly trains agents' policies from preference data, without relying on an explicit reward model. Our main contributions are as follows:

First, we introduce a new algorithm, O-MAPL (Offline Multi-Agent Preference Learning) for multi-agent PbRL. O-MAPL exploits the inherent relationship between the reward and the soft-Q functions in MaxEnt RL (Garg et al., 2021; 2023) to directly learn the soft Q-function from preference data, rather than recovering the reward function explicitly. Once the Q-function is learned, the optimal policy can be derived. This one-phase learning process is carried out under the centralized training with decentralized execution (CTDE) paradigm (Oliehoek et al., 2008; Kraemer & Banerjee, 2016), allowing effective training of local policies.

Implementing this end-to-end process within the CTDE framework is far from being trivial. It requires appropriate mixing networks for value factorization to preserve the convexity of the preference-based learning objective and ensure local-global consistency in policy optimality. As a second contribution, we introduce a simple yet effective value factorization method and provide a comprehensive theoretical analysis of the convexity and local-global consistency requirements. This approach enables stable and efficient policy training.

Finally, we conduct extensive experiments on two benchmarks, SMAC and MAMuJoCo, using preference data generated by both rule-based and large language model approaches. The results show that our O-MAPL consistently outperforms existing methods across various tasks.

## 2. Related Work

**Offline multi-agent reinforcement learning (MARL).** Our work is related to offline MARL, relying solely on offline data to learn policies without direct interaction with the environment. Unlike standard offline MARL, we consider data only showing pairwise trajectory preferences (without rewards). Like offline MARL, it faces challenges such as distributional shift and complex interactions in large joint state and action spaces. Many existing MARL methods use the CTDE framework (Oliehoek et al., 2008; Kraemer & Banerjee, 2016), enabling efficient learning while allowing independent operation of agents. Regularization techniques are also applied to mitigate distributional shift (Yang et al., 2021; Pan et al., 2022; Shao et al., 2024; Wang et al., 2022).

For example, some works extend CQL (Kumar et al., 2020), a well-known single-agent offline RL algorithm, to multi-agent settings (Pan et al., 2022; Shao et al., 2024). Others adopt the popular DICE framework, which regulates policies in the occupancy space to address out-of-distribution (OOD) issues in both competitive and cooperative settings (Matsunaga et al., 2023; Bui et al., 2025). Additionally, (Wang et al., 2022) explore a policy constraint framework to tackle OOD problems. Some studies apply sequence modeling techniques to solve offline MARL using supervised learning

approaches (Meng et al., 2023; Tseng et al., 2022).

**Preference-based reinforcement learning (PbRL).** Early works developed general frameworks using linear approximations or Bayesian models to incorporate human feedback on policies, trajectories, and state/action pairwise comparisons into policy learning (Fürnkranz et al., 2012; Akrour et al., 2012; 2011; Wilson et al., 2012). Recent studies have shown the effectiveness of training deep neural networks in complex domains with thousands of preference queries, typically following a two-phase approach: first, supervised learning to train a reward model, then RL to optimize the policy. For example, (Christiano et al., 2017) uses the Bradley-Terry model for pairwise preferences and methods like A2C (Mnih, 2016) to refine the policy. Subsequent studies have expanded this framework to scenarios like preference elicitation (Mukherjee et al., 2024; Lee et al., 2021), few-shot learning (Hejna III & Sadigh, 2023), data and preference augmentation (Ibarz et al., 2018; Zhang et al., 2023), list-wise learning (Choi et al., 2024), hindsight preference learning (Gao et al., 2024), and Transformer-based learning (Kim et al., 2023).

Training reward models aligned with human preferences can be costly, requiring large volumes of preference data, especially in complex domains. This has led to a shift towards end-to-end frameworks that directly learn optimal policies from preference data, bypassing explicit reward models. For example, (Hejna et al., 2023) and (An et al., 2023) use contrastive learning to eliminate reward modeling, while (Kang et al., 2023) employs information matching to learn optimal policies in one step. In (Hejna & Sadigh, 2024), the IPL algorithm learns a Q-function directly from expert preferences, instead of modeling the reward function.

While preference-based RL is well-explored in single-agent settings, research in multi-agent settings remains limited due to the complexity of agent interactions and large joint state-action spaces. Only a few studies have extended the two-phase preference-based framework to multi-agent settings (Kang et al., 2024; Zhang et al., 2024a). Building on IPL's success in single-agent settings (Hejna & Sadigh, 2024), we leverage the reward-Q-function relationship to avoid explicit reward modeling. Adapting this to multi-agent environments is challenging, requiring careful design of mixing networks within the CTDE framework and a thorough theoretical analysis of the preference-based learning objective's convexity and global-local policy consistency.

## 3. Background

**Multi-agent Reinforcement Learning.** We focus on cooperative MARL, modeled as a multi-agent Partially Observable Markov Decision Process (POMDP) defined by $\mathcal{M} = \langle \mathcal{S}, \mathcal{A}, P, r, \mathcal{Z}, \mathcal{O}, n, \mathcal{N}, \gamma \rangle$ where $n$ is the number of

agents, and $\mathcal{N} = \{1, \ldots, n\}$ is the set of agents. The true state of the environment is denoted by $\mathbf{s} \in \mathcal{S}$, and the joint action space is given by $\mathcal{A} = \prod_{i \in \mathcal{N}} \mathcal{A}_i$, where $\mathcal{A}_i$ is the set of actions for agent $i \in \mathcal{N}$. At each time step, every agent $i \in \mathcal{N}$ selects an action $a_i \in \mathcal{A}_i$, resulting in a joint action $\mathbf{a} = (a_1, a_2, \ldots, a_n) \in \mathcal{A}$. The transition dynamics are described by $P(\mathbf{s}'|\mathbf{s}, \mathbf{a}) : \mathcal{S} \times \mathcal{A} \times \mathcal{S} \to [0, 1]$, which is the probability of moving to the next state $\mathbf{s}'$ given the current state $\mathbf{s}$ and joint action $\mathbf{a}$. The discount factor $\gamma \in [0, 1)$ determines the relative importance of future rewards.

In partial observability settings, each agent receives a local observation $o_i \in \mathcal{O}_i$ based on the function $\mathcal{Z}_i(\mathbf{s}) : \mathcal{S} \to \mathcal{O}_i$, and the joint observation is denoted by $\mathbf{o} = (o_1, o_2, \ldots, o_n)$. In cooperative MARL, agents share a global reward function $r(\mathbf{s}, \mathbf{a}) : \mathcal{S} \times \mathcal{A} \to \mathbb{R}$. The objective is to learn a joint policy $\pi_{\text{tot}} = \{\pi_1, \ldots, \pi_n\}$ that maximizes the expected discounted cumulative rewards $\mathbb{E}_{(\mathbf{o}, \mathbf{a}) \sim \pi_{\text{tot}}} [\sum_{t=0}^{\infty} \gamma^t r(\mathbf{s}_t, \mathbf{a}_t)]$. In offline settings, a dataset $\mathcal{D}$ is pre-collected by sampling from a behavior policy $\mu_{\text{tot}} = \{\mu_1, \ldots, \mu_n\}$. Policy learning is then carried out using this dataset $\mathcal{D}$ only.

**MaxEnt Reinforcement Learning.** Standard RL optimizes a policy that maximizes the expected discounted cumulative rewards $\mathbb{E}_{\pi_{\text{tot}}} [\sum_{t=0}^{\infty} \gamma^t r(\mathbf{s}_t, \mathbf{a}_t)]$[1], where $(\mathbf{s}_t, \mathbf{a}_t)$ are sampled at each time step $t$ from the trajectory distribution induced by the joint policy $\pi_{\text{tot}}$. In a generalized MaxEnt RL, the standard reward objective is augmented with a KL-divergence term between the joint policy and a behavior $\mu_{\text{tot}}$ that generates the offline dataset, as follows:

$$\mathbb{E}_{\pi_{\text{tot}}} \left[ \sum_{t=0}^{\infty} \gamma^t \left( r(\mathbf{s}_t, \mathbf{a}_t) - \beta \log \frac{\pi_{\text{tot}}(\mathbf{a}_t|\mathbf{s}_t)}{\mu_{\text{tot}}(\mathbf{a}_t|\mathbf{s}_t)} \right) \right],$$

where $\beta$ is the regularization parameter. Setting $\mu_{\text{tot}}$ to the uniform distribution reduces this to the standard MaxEnt RL objective. The regularization term enforces a conservative KL constraint, keeping the learned policy close to the behavior policy and addressing offline RL's out-of-distribution challenges (Haarnoja et al., 2018; Neu et al., 2017).

In the above MaxEnt framework, the soft-Bellman operator $\mathcal{B}^* : \mathbb{R}^{\mathcal{S} \times \mathcal{A}} \to \mathbb{R}^{\mathcal{S} \times \mathcal{A}}$ is defined as $(\mathcal{B}_r^* Q_{\text{tot}})(\mathbf{s}, \mathbf{a}) = r(\mathbf{s}, \mathbf{a}) + \gamma \mathbb{E}_{\mathbf{s}' \sim P(\cdot|\mathbf{s}, \mathbf{a})} V_{\text{tot}}(\mathbf{s}')$, where $Q_{\text{tot}}$ is the soft-global-Q function and $V_{\text{tot}}$ is the optimal soft-global-value function computed as a log-sum-exp of $Q_{\text{tot}}$, as follows:

$$V_{\text{tot}}(\mathbf{s}) = \beta \log \left[ \sum_{\mathbf{a} \sim \mu_{\text{tot}}(\cdot|\mathbf{s})} \mu_{\text{tot}}(\mathbf{a}|\mathbf{s}) \exp \left( \frac{Q_{\text{tot}}(\mathbf{s}, \mathbf{a})}{\beta} \right) \right]$$

The Bellman equation $(\mathcal{B}_r^* Q_{\text{tot}}) = Q_{\text{tot}}$ will yield a unique optimal global Q-function $Q_{\text{tot}}^*$ and the corresponding optimal policy is given by (Haarnoja et al., 2018):

$$\pi_{\text{tot}}^*(\mathbf{a}|\mathbf{s}) = \mu_{\text{tot}}(\mathbf{a}|\mathbf{s}) \exp \left( \frac{Q_{\text{tot}}^*(\mathbf{s}, \mathbf{a}) - V_{\text{tot}}^*(\mathbf{s})}{\beta} \right). \quad (1)$$

---

[1]We adapt the formulas from single-agent MaxEnt RL to the multi-agent setting, ensuring consistency in notation.

where $V_{\text{tot}}^*$ is the log-sum-exp of $Q_{\text{tot}}^*$. Moreover, by rearranging the Bellman equation, we get the so-called inverse soft Bellman-operator, formulated as follows:

$$(\mathcal{T}^* Q_{\text{tot}})(\mathbf{s}, \mathbf{a}) = Q_{\text{tot}}(\mathbf{s}, \mathbf{a}) - \gamma \mathbb{E}_{\mathbf{s}' \sim P(\cdot|\mathbf{s}, \mathbf{a})} V_{\text{tot}}(\mathbf{s}')$$

An important observation here is the one-to-one mapping between any $Q_{\text{tot}}$ and $r(\mathbf{s}, \mathbf{a})$, i.e., $r(\mathbf{s}, \mathbf{a}) = (\mathcal{T}^* Q_{\text{tot}})(\mathbf{s}, \mathbf{a})$. This property has been extensively utilized in inverse RL (Garg et al., 2021; Hejna & Sadigh, 2024; Bui et al., 2024). The key idea is that, rather than explicitly recovering a reward function, the unique mapping enables the reformulation of reward learning as a Q-learning problem. This approach improves stability and can directly recover the optimal policy from the learned Q-function using (1).

Note that, in POMDP scenarios, the global state $\mathbf{s}$ is not directly accessible during training and is instead represented by the joint observations $\mathbf{o}$ from the agents. For notational convenience, we use the global state $\mathbf{s}$ in our formulation; however, in practice, it corresponds to the joint observation $\mathcal{Z}(\mathbf{s})$. Specifically, terms like $\pi_{\text{tot}}(\mathbf{s}, \mathbf{a})$ and $Q_{\text{tot}}(\mathbf{s}, \mathbf{a})$ actually refer to $\mu_{\text{tot}}(\mathbf{o}, \mathbf{a})$ and $Q_{\text{tot}}(\mathbf{o}, \mathbf{a})$, where $\mathbf{o} = \mathcal{Z}(\mathbf{s})$.

## 4. Multi-agent Preference-based RL (PbRL)

### 4.1. Preference-based Inverse Q-learning

Following prior works (Christiano et al., 2017; Lee et al., 2021; Kang et al., 2024), we assume access to pairwise preference data. The data, collected from humans (or experts), consists of pairs of trajectories $(\sigma_1, \sigma_2)$, where $\sigma_1$ is preferred over $\sigma_2$. Each trajectory $\sigma$ is a sequence of joint (state, action) pairs: $\sigma = \{(\mathbf{s}_1, \mathbf{a}_1), \ldots, (\mathbf{s}_K, \mathbf{a}_K)\}$. Let $\mathcal{P}$ denote the preference dataset, comprising several pairwise comparisons $(\sigma_1, \sigma_2)$. The goal of PbRL is to recover the underlying reward function and expert policies from $\mathcal{P}$.

A common approach in PbRL is to model the expert's preferences using the simple and intuitive Bradley-Terry model (Bradley & Terry, 1952), which computes the probability of the expert preferring $\sigma_1$ over $\sigma_2$ (denoted as $\sigma_1 \succ \sigma_2$) as:

$$P(\sigma_1 \succ \sigma_2) = \frac{e^{\sum_{(\mathbf{s}, \mathbf{a}) \in \sigma_1} r_E(\mathbf{s}, \mathbf{a})}}{e^{\sum_{(\mathbf{s}, \mathbf{a}) \in \sigma_1} r_E(\mathbf{s}, \mathbf{a})} + e^{\sum_{(\mathbf{s}, \mathbf{a}) \in \sigma_2} r_E(\mathbf{s}, \mathbf{a})}}$$

where $r_E(\mathbf{s}, \mathbf{a})$ is the reward function of the expert. Using this model, a direct approach to recovering the expert reward function $r_E$ involves maximizing the likelihood of the preference data $\mathcal{P}$, which can be formulated as follows:

$$\max_{r_E} \mathcal{L}(r_E|\mathcal{P}) = \max_{r_E} \sum_{(\sigma_1, \sigma_2) \in \mathcal{P}} \ln P(\sigma_1 \succ \sigma_2)$$

Once the expert rewards $r_E$ are recovered, a policy can be learned by training a MARL algorithm. This method is referred to as a two-phase approach, where the reward learning and policy optimization are performed separately.

The MaxEnt RL framework discussed above provides an alternative approach to integrate reward and policy recovery into a single learning process. This is achieved by leveraging the unique mapping between a reward function and a Q-function. Multi-agent PbRL is thereby transformed into the Q-space, where the preference probability over a trajectory pair $(\sigma_1, \sigma_2)$ can be computed as follows:

$$P(\sigma_1 \succ \sigma_2 | Q_{tot}) = \frac{e^{\sum_{\sigma_1}(\mathcal{T}^* Q_{tot})(\mathbf{s}, \mathbf{a})}}{e^{\sum_{\sigma_1}(\mathcal{T}^* Q_{tot})(\mathbf{s}, \mathbf{a})} + e^{\sum_{\sigma_2}(\mathcal{T}^* Q_{tot})(\mathbf{s}, \mathbf{a})}}$$

After solving the maximum likelihood problem, the derived $Q_{\text{tot}}$ and $V_{\text{tot}}$ can be used directly to recover a policy via the soft policy formula (1), eliminating the need for an additional MARL algorithm. This unified, single-phase approach integrates reward and policy learning, streamlining the process. It enhances training stability and consistency by reducing discrepancies that arise from separate reward and policy learning. This approach also mitigates issues like error propagation and misalignment between the reward function and policy optimization. Training in the Q-space has been shown to outperform training in the reward space.

### 4.2. Value Factorization

The training objective in the Q-space can be formulated as:

$$\max_{Q_{\text{tot}}} \mathcal{L}(Q_{\text{tot}} | \mathcal{P}) = \max_{Q_{\text{tot}}} \sum_{(\sigma_1, \sigma_2) \in \mathcal{P}} \ln P(\sigma_1 \succ \sigma_2 | Q_{\text{tot}})$$

While this objective works in single-agent settings, applying it to multi-agent scenarios is challenging due to the large state and action spaces. To address this, we apply value factorization in the CTDE framework. However, solving PbRL under CTDE is complex, as the objective involves several components tied to $Q_{\text{tot}}$ and $V_{\text{tot}}$. Thus, a carefully designed value factorization method is needed to ensure consistency between global and local policies.

To address these challenges, we propose a value factorization method, specifically designed to ensure scalability in multi-agent environments while preserving the alignment between global and local objectives, thereby enabling stable and effective learning. Our approach involves factorizing the global value functions $Q_{\text{tot}}$ and $V_{\text{tot}}$ into local functions using a mixing network architecture. Specifically, let $\mathbf{q}(\mathbf{s}, \mathbf{a}) = \{q_1(s_1, a_1), \ldots, q_n(s_n, a_n)\}$ be a set of local Q-functions, and $\mathbf{v}(\mathbf{s}) = \{v_1(s_1), \ldots, v_n(s_n)\}$ represent a set of local V-functions. To enable centralized learning, we introduce a mixing network $\mathcal{M}_w$, parameterized by learnable weights $w$, which combines the local functions $\mathbf{q}$ and $\mathbf{v}$ to construct the global value functions $Q_{\text{tot}}$ and $V_{\text{tot}}$ as follows:

$$Q_{\text{tot}}(\mathbf{s}, \mathbf{a}) = \mathcal{M}_w[\mathbf{q}(\mathbf{s}, \mathbf{a})]; \quad V_{\text{tot}}(\mathbf{s}) = \mathcal{M}_w[\mathbf{v}(\mathbf{s})].$$

For notational simplicity, let us define:

$$\begin{aligned} R_w[\mathbf{q}, \mathbf{v}](\mathbf{s}, \mathbf{a}) &= Q_{tot}(\mathbf{s}, \mathbf{a}) - \gamma \mathbb{E}_{\mathbf{s}' \sim P(\cdot | \mathbf{s}, \mathbf{a})} V_{tot}(\mathbf{s}') \\ &= \mathcal{M}_w[\mathbf{q}(\mathbf{s}, \mathbf{a})] - \gamma \mathbb{E}_{\mathbf{s}' \sim P(\cdot | \mathbf{s}, \mathbf{a})} \mathcal{M}_w[\mathbf{v}(\mathbf{s}')] \end{aligned}$$

The mixing function $\mathcal{M}_w$ can be either a linear combination (single-layer) or a nonlinear combination (e.g., a two-layer network with ReLU activation). Our work uses the simple linear structure, which has two key advantages over the non-linear approach. **First**, a two-layer structure often causes over-fitting and poor performance, especially in offline settings with limited data (Bui et al., 2025). **Second**, the linear structure ensures convexity in the learning objectives within the Q-space, leading to stable optimization and consistent training—benefits not present under a two-layer mixing network structure.

Overall, the training objective function, under the described mixing architecture, can be now expressed as follows:

$$\begin{aligned} \mathcal{L}(\mathbf{q}, \mathbf{v}, w) = \sum_{(\sigma_1, \sigma_2) \in \mathcal{P}} & \sum_{(\mathbf{s}, \mathbf{a}) \in \sigma_1} R_w[\mathbf{q}, \mathbf{v}](\mathbf{s}, \mathbf{a}) \\ & - \log \left( e^{\sum_{\sigma_1} R_w[\mathbf{q}, \mathbf{v}](\mathbf{s}, \mathbf{a})} + e^{\sum_{\sigma_2} R_w[\mathbf{q}, \mathbf{v}](\mathbf{s}, \mathbf{a})} \right) \\ & + \phi(R_w[\mathbf{q}, \mathbf{v}](\mathbf{s}, \mathbf{a})) \end{aligned}$$

where $\phi(\cdot)$ is a concave regularization function used to prevent unbounded reward functions. In our experiments we choose a $\chi^2$ regularizer of the form $\phi(x) = -\frac{1}{2}x^2 + x$, which is also a commonly used regularizer in prior works.

It is important to note that $Q_{\text{tot}}$ and $V_{\text{tot}}$ must satisfy the Bellman operator, meaning that $V_{\text{tot}}$ needs to be the log-sum-exp of $Q_{\text{tot}}$. To achieve this, we train $\mathcal{M}_w[\mathbf{v}(\mathbf{s})]$ (or $V_{tot}$) to approximate the log-sum-exp formulation:

$$V_{tot}(\mathbf{s}) = \beta \log \left( \sum_{\mathbf{a} \in \mathcal{A}} \mu_{\text{tot}}(\mathbf{a}|\mathbf{s}) e^{Q_{tot}(\mathbf{s}, \mathbf{a})/\beta} \right),$$

However, this can become computationally impractical in certain scenarios, such as environments with continuous action spaces. To address this, Extreme Q-Learning (XQL) (Garg et al., 2023) provides an efficient method to update the $V$-function. Specifically, we define the *extreme-V loss objective* under our mixing framework as follows:

$$\begin{aligned} \mathcal{J}(\mathbf{v}) = \mathbb{E}_{(\mathbf{s}, \mathbf{a}) \sim \mu_{\text{tot}}} & \left[ e^{\frac{\mathcal{M}_w[\mathbf{q}(\mathbf{s}, \mathbf{a})] - \mathcal{M}_w[\mathbf{v}(\mathbf{s})]}{\beta}} \right] \\ & - \mathbb{E}_{(\mathbf{s}, \mathbf{a}) \sim \mu_{\text{tot}}} \left[ \frac{\mathcal{M}_w[\mathbf{q}(\mathbf{s}, \mathbf{a})] - \mathcal{M}_w[\mathbf{v}(\mathbf{s})]}{\beta} \right] - 1. \end{aligned}$$

Minimizing $\mathcal{J}(\mathbf{v})$ over $\mathbf{v}$ ensures that $\mathcal{M}_w[\mathbf{v}(\mathbf{s})]$ converges to the log-sum-exp value (Garg et al., 2023):

$$\mathcal{M}_w[\mathbf{v}(\mathbf{s})] = \beta \log \left( \sum_{\mathbf{a} \in \mathcal{A}} \mu_{\text{tot}}(\mathbf{a}|\mathbf{s}) e^{\mathcal{M}_w[\mathbf{q}(\mathbf{s}, \mathbf{a})]/\beta} \right).$$

Following this approach, training the local functions $\mathbf{q}$ and $\mathbf{v}$ can proceed through the following alternating updates:

- **Update $\mathbf{q}$, $w$**: Maximize $\mathcal{L}(\mathbf{q}, \mathbf{v}, w)$, the likelihood objective for preference learning.

- **Update $\mathbf{v}$**: Minimize the extreme-V loss $\mathcal{J}(\mathbf{v})$ to enforce consistency with the log-sum-exp equation.

The following proposition shows that the learning objective functions under our mixing architectures possess appealing properties, which contribute to stable and robust training.

**Proposition 4.1** (Convexity). *The loss $\mathcal{L}(\mathbf{q}, \mathbf{v}, w)$ is concave in $\mathbf{q}$ and $w$ (the parameters of the mixing networks), while the extreme-V loss function $\mathcal{J}(\mathbf{v})$ is convex in $\mathbf{v}$.*

Given that the objective is to maximize the likelihood function $\mathcal{L}(\mathbf{q}, \mathbf{v}, w)$ and minimize the extreme-V function $\mathcal{J}(\mathbf{v})$, the concavity of $\mathcal{L}$ in $\mathbf{q}$ and $w$, and the convexity of $\mathcal{J}$ in $\mathbf{v}$, guarantees unique convergence (theoretically) within the $\mathbf{q}$ and $\mathbf{v}$ spaces, ensures a stable training process in practice.

It is important to note that convexity is guaranteed only under single-layer mixing structures, where $\mathcal{M}_w[\cdot]$ is linear in its inputs. This result is formalized below:

**Proposition 4.2** (Non-convexity under two-layer mixing networks). *If the mixing networks $\mathcal{M}_w[\mathbf{q}]$ and $\mathcal{M}_w[\mathbf{v}]$ are two-layer (or multi-layer) feed-forward networks, the preference-based loss function $\mathcal{L}(\mathbf{q}, \mathbf{v}, w)$ is no longer concave in $\mathbf{q}$ or $w$, and the extreme-V loss function $\mathcal{J}(\mathbf{v})$ is **not** convex in $\mathbf{v}$.*

While two-layer feed-forward mixing networks have been employed in several prior online MARL works, single-layer mixing networks (i.e., linear combinations) have been favored in recent offline MARL works (Wang et al., 2022; Bui et al., 2025). It was demonstrated that using a two-layer network can lead to over-fitting issues, resulting in worse performance compared to their single-layer counterparts (Bui et al., 2025). The results in Prop. 4.2 further suggest that, in offline preference-based learning, a single-layer setup is more efficient and better suited to achieve robust and stable performance.

### 4.3. Local Policy Extraction

**Simple local-value-based extraction approach.** Globally optimal policies can be extracted from $Q_{\text{tot}}$ and $V_{\text{tot}}$ based on (1). For decentralized execution, local policies can be derived from local values similarly (Wang et al., 2022):

$$\pi_i^*(a_i|s_i) = \mu_i(a_i|s_i) \exp\left( \frac{w_i^q q_i(s_i, a_i) - w_i^v v_i(s_i)}{\beta} \right), \quad (2)$$

where $w_i^q$ and $w_i^v$ are the weights of the mixing function $\mathcal{M}_w[\mathbf{q}]$ and $\mathcal{M}_w[\mathbf{v}]$, and $\mu_i(\cdot)$ are the local behavior policies. Assuming the behavior policy is decomposable into local components, i.e., $\mu_{\text{tot}}(\mathbf{a}|\mathbf{s}) = \prod_i \mu_i(a_i|s_i)$, this policy extraction method guarantees *global-local consistency (GLC)* — ensuring alignment between the optimal global and local policies — such that $\pi_{\text{tot}}^*(\mathbf{a}|\mathbf{s}) = \prod_i \pi_i^*(a_i|s_i)$.

This approach has been used in prior work but has notable limitations. First, GLC holds only with a linear mixing structure; a two-layer feed-forward network breaks this property. Second, policies recovered from (2) may not be feasible, as

the sum of $\pi_i^*(a_i|s_i)$ over all $a_i$ might not equal one. To ensure feasibility, normalization is required, but it disrupts the GLC principle, breaking consistency between global and local policies. Also, the local functions $v_i$ and $q_i$ may not satisfy the local Bellman equality (i.e., $v_i$ is not guaranteed to be the log-sum-exp of $q_i$), causing the soft policy formula to misalign with MaxEnt RL principles at the local level.

**Our weighted behavior cloning approach.** We propose an alternative approach that offers several advantages over the previous method. Our policy extraction is based on BC, a technique commonly used in offline RL algorithms (Garg et al., 2023; Bui et al., 2025). This approach preserves the GLC property and ensures that the extracted local policies are valid, even with nonlinear mixing structures.

In general, the global policy can be extracted by solving the following weighted behavior cloning (WBC) problem:

$$\max_{\pi_{\text{tot}} \in \Pi_{\text{tot}}} \left\{ \mathbb{E}_{\mathbf{s}, \mathbf{a} \sim \mu_{\text{tot}}} \left[ e^{\frac{Q_{\text{tot}}(\mathbf{s}, \mathbf{a}) - V_{\text{tot}}(\mathbf{s})}{\beta}} \log \pi_{\text{tot}}(\mathbf{a}|\mathbf{s}) \right] \right\}, \quad (3)$$

where $\Pi_{\text{tot}}$ represents the feasible set of global policies. Here, we assume that $\Pi_{\text{tot}}$ contains decomposable global policies, i.e., $\Pi_{\text{tot}} = \{\pi_{\text{tot}} \mid \exists \pi_i, \forall i \in \mathcal{N}, \text{ such that } \pi_{\text{tot}}(\mathbf{a}|\mathbf{s}) = \prod_{i \in \mathcal{N}} \pi_i(a_i|s_i)\}$. In other words, $\Pi_{\text{tot}}$ consists of global policies that can be expressed as a product of local policies. This decomposability is highly useful for decentralized learning and has been widely adopted in multi-agent reinforcement learning (MARL) (Wang et al., 2022; Bui et al., 2024; Zhang et al., 2021).

While solving (3) can explicitly recover an optimal global policy and is practical via sampling $(\mathbf{s}, \mathbf{a})$ from the data, it does not support the learning of local policies, which is essential under the CTDE principle. To address this, we propose solving the following local WBC problem:

$$\max_{\pi_i} \left\{ \mathbb{E}_{\mathbf{s}, \mathbf{a} \sim \mu_{\text{tot}}} \left[ e^{\frac{Q_{\text{tot}}(\mathbf{s}, \mathbf{a}) - V_{\text{tot}}(\mathbf{s})}{\beta}} \log \pi_i(a_i|s_i) \right] \right\} \quad (4)$$

The local WBC approach has several key advantages. First, the weighting term $e^{\frac{Q_{\text{tot}}(\mathbf{s}, \mathbf{a}) - V_{\text{tot}}(\mathbf{s})}{\beta}}$ directly influences local policy optimization and is computed from global observations and actions. This ensures local policies are optimized with global information, maintaining consistency in cooperative multi-agent systems. Furthermore, as shown in Theorem 4.3, optimizing local policies via WBC always results in valid policies that align with the global WBC objective, preserving global-local consistency (GLC). Importantly, these benefits hold regardless of the mixing structure (e.g., 1-layer or 2-layer networks), offering significant advantages over the local-value-extraction method.

**Theorem 4.3** (Global-Local Consistency (GLC)). *Let $\pi_i^*$ be the optimal solution to the local WBC problem in (4). Then, the global policy $\pi_{\text{tot}}^*$, defined as $\pi_{\text{tot}}^*(\mathbf{s}, \mathbf{a}) = \prod_i \pi_i^*(a_i|s_i)$, is also optimal for the global WBC problem in (3).*

We next formally express the relationship between recovered local policies and value functions. We assume that the behavior policy is decomposable, i.e., $\mu_{tot}(\mathbf{a}|\mathbf{a}) = \prod_i \mu_i(a_i|s_i)$, and the mixing structures are defined as $\mathcal{M}_w[\mathbf{q}(\mathbf{s}, \mathbf{a})] = \sum_i w_i^q q_i(s_i, a_i) + b_q$ and $\mathcal{M}_w[\mathbf{v}(\mathbf{s})] = \sum_i w_i^v v_i(s_i) + b_v$. This relationship is formalized in the following theorem:

**Theorem 4.4.** *Let $\pi_i^*$ be optimal to the local WBC, then the following equality holds for all $s_i \in \mathcal{S}_i, a_i \in \mathcal{A}_i$:*

$$\pi_i^*(a_i|s_i) = \frac{\eta(s_i)}{\Delta(s_i)} \mu_i(a_i|s_i) e^{\frac{w_i^q q_i(s_i, a_i) - w_i^v v_i(s_i)}{\beta}} \quad (5)$$

*where $\eta(s_i)/\Delta(s_i)$ are correction terms.*[2]

Theorem 4.4 highlights key aspects of our approach. First, as seen in (2), directly computing local policies from the local value functions $q_i$ and $v_i$ alone may yield invalid policies that don't form proper probability distributions. The term $\eta(s_i)/\Delta(s_i)$ in (9) acts as a correction factor, normalizing the policies to ensure $\sum_{a_i} \pi_i^*(a_i|s_i) = 1$. Furthermore, the proof of Theorem 4.4 shows that both $\eta(s_i)/\Delta(s_i)$ and the local policy $\pi_i^*(a_i|s_i)$ depend on the value functions of other agents. This dependency supports the principle of credit assignment in cooperative MARL, ensuring each agent's policy accounts for the actions and rewards of others.

Additionally, while $V_{tot}$ is the log-sum-exp of $Q_{tot}$, this might not be the case for the local $v_i$ and $q_i$ functions. The following proposition demonstrates that $v_i$ can indeed be expressed as a log-sum-exp of $q_i$, along with *an additional term* that depends on the local functions of other agents.

**Proposition 4.5.** *Each local value $v_i$ can be expressed as a (modified) log-sum-exp of the local Q-function $q_i$:*

$$v_i(s_i) = \frac{\beta}{w_i^v} \log \sum_{a_i \sim \mu_i(\cdot|s_i)} e^{\frac{w_i^q}{\beta} q_i(s_i, a_i)} + \frac{\beta}{w_i^v} \log\left(\frac{\eta(s_i)}{\Delta(s_i)}\right)$$

Prop. 4.5 indicates that $v_i(s_i)$ is also determined by a log-sum-exp of $q_i(s_i, a_i)$ with an additional term $\log\left(\frac{\eta(s_i)}{\Delta(s_i)}\right)$.

## 5. Practical Algorithm

In the context of POMDPs, we do not have direct access to the global states. To better reflect the practical aspects, we change the notation of global states used previously to global observations. For example, the local value function is now defined as a function of local observations, $v_i(o_i)$.

We construct a local Q-value network $q_i(o_i, a_i|\psi_q)$ and a local value network $v_i(o_i|\psi_v)$, where $\psi_q$ and $\psi_v$ are learnable parameters. The global Q and V functions are then aggregated using two mixing networks with a shared set of learnable parameters $\theta$, formulated as follows:

$$V_{tot}(\mathbf{o}) = \mathcal{M}_\theta[\mathbf{v}(\mathbf{o}|\psi_v)]; \quad Q_{tot}(\mathbf{o}, \mathbf{a}) = \mathcal{M}_\theta[\mathbf{q}(\mathbf{o}, \mathbf{a}|\psi_q)],$$

[2]Detailed formulations of these terms are in Appendix A.4.

---

**Algorithm 1 O-MAPL**

1: **Input:** Parameters $\theta, \psi_q, \psi_v, \omega_i$. Offline data $\mathcal{P}$.
2: **Output:** Local optimized polices $\pi_i$.
3: **for** *a certain number of training steps* **do**
4:     Update $\psi_q$ and $\theta$ to maximize $\mathcal{L}(\psi_q, \psi_v, \theta)$
5:     Update $\psi_v$ to minimize the Extreme-V $J(\psi_v)$
6:     Update $\omega_i$ to maximize the local WBC loss $\Psi(\omega_i)$
7: **end for**
8: Return $\pi_i(a_i|o_i; \omega_i), i = 1, ..., n$

---

where $\mathcal{M}_\theta[\cdot]$ is a linear combination (or a one-layer mixing network) of its inputs with non-negative weights:

$$\mathcal{M}_\theta[\mathbf{v}(\mathbf{o}|\psi_v)] = \mathbf{v}(\mathbf{o}|\psi_v)^\top W_\theta^{\mathbf{o}} + b_\theta^{\mathbf{o}} \quad (6)$$

$$\mathcal{M}_\theta[\mathbf{q}(\mathbf{o}, \mathbf{a}|\psi_q)] = \mathbf{q}(\mathbf{o}, \mathbf{a}|\psi_q)^\top W_\theta^{\mathbf{o},\mathbf{a}} + b_\theta^{\mathbf{o},\mathbf{a}}, \quad (7)$$

Here, $W_\theta^{\mathbf{o}}, b_\theta^{\mathbf{o}}, W_\theta^{\mathbf{o},\mathbf{a}}, b_\theta^{\mathbf{o},\mathbf{a}}$ are the weights of the mixing networks, modeled as hyper-networks that take the global observation $\mathbf{o}$, joint action $\mathbf{a}$, and the learnable parameters $\theta$ as inputs. In this setup, we employ the same mixing network $\mathcal{M}_\theta$ to combine both the local $V$ and $Q$ functions, ensuring consistency and scalability in the aggregation process.

The practical training objective function for the local Q functions can be calculated as:

$$\mathcal{L}(\psi_q, \psi_v, \theta) = \sum_{(\sigma_1,\sigma_2)\in\mathcal{P}} \sum_{(\mathbf{o},\mathbf{a},\mathbf{o}')\in\sigma_1} R(\mathbf{o}, \mathbf{a}, \mathbf{o}')$$
$$- \log\left(e^{\sum_{\sigma_1} R(\mathbf{o},\mathbf{a},\mathbf{o}')} + e^{\sum_{\sigma_2} R(\mathbf{o},\mathbf{a},\mathbf{o}')}\right) + \sum_{\mathcal{P}} \phi(R(\mathbf{o}, \mathbf{a}, \mathbf{o}'))$$

where $R(\mathbf{o}, \mathbf{a}) = \mathcal{M}_\theta[\mathbf{q}(\mathbf{o}, \mathbf{a}|\psi_q)] - \gamma\mathcal{M}_\theta[\mathbf{v}(\mathbf{o}'|\psi_v)]$. Moreover, the extreme-V can be practically estimated as:

$$\mathcal{J}(\psi_v) = \mathbb{E}_{(\mathbf{o},\mathbf{a})\sim\mathcal{P}}\left[e^{\frac{\mathcal{M}_\theta[\mathbf{q}(\mathbf{o},\mathbf{a}|\psi_q)] - \mathcal{M}_\theta[\mathbf{v}(\mathbf{o}|\psi_v)]}{\beta}}\right]$$
$$- \mathbb{E}_{(\mathbf{o},\mathbf{a})\sim\mathcal{P}}\left[\frac{\mathcal{M}_\theta[\mathbf{q}(\mathbf{o},\mathbf{a}|\psi_q)] - \mathcal{M}_\theta[\mathbf{v}(\mathbf{o}|\psi_v)]}{\beta}\right] - 1$$

For the policy extraction, let $\pi_i(a_i|o_i; \omega_i)$ be a local policy network for each agent $i$, where $\omega$ are learnable parameters. We update the local policies using the following local WBC:

$$\Psi(\omega_i) = \sum_{\mathbf{o},\mathbf{a}\sim\mathcal{P}}\left[e^{\frac{\mathcal{M}_\theta[\mathbf{q}(\mathbf{o},\mathbf{a}|\psi_q)] - \mathcal{M}_\theta[\mathbf{v}(\mathbf{o}|\psi_v)]}{\beta}} \log \pi_i(o_i|s_i; \omega_i)\right]$$

The outline of our O-MAPL is shown in Algorithm 1.

## 6. Experiments

We evaluate the performance of our O-MAPL in different complex MARL environments, including: multi-agent StarCraft II (i.e., SMACv1 (Samvelyan et al., 2019), SMACv2 (Ellis et al., 2022)) and multi-agent Mujoco (de Witt et al., 2020a) benchmarks. Detailed descriptions of these benchmarks are in the appendix.

| Tasks | Rule-based | | | | | LLM-based | | | | |
|---|---|---|---|---|---|---|---|---|---|---|
| | BC | IIPL | IPL-VDN | SL-MARL | O-MAPL (ours) | BC | IIPL | IPL-VDN | SL-MARL | O-MAPL (ours) |
| 2c_vs_64zg | 59.6±25.0 | 60.4±24.7 | 71.1±22.0 | 63.5±24.0 | **74.4±24.7** | 65.6±24.6 | 60.2±25.9 | 77.0±21.3 | 65.2±21.2 | **79.5±19.6** |
| 5m_vs_6m | 16.8±18.0 | 14.3±17.0 | 16.8±18.0 | 16.0±18.9 | **19.3±19.6** | 18.2±18.4 | 15.0±17.5 | 18.0±19.2 | 17.4±19.4 | **20.7±20.5** |
| 6h_vs_8z | 0.6±3.8 | 0.2±2.2 | 2.5±7.6 | 1.6±6.8 | **4.5±11.0** | 0.8±4.3 | 0.4±3.1 | 3.5±9.2 | 3.7±8.9 | **6.1±11.2** |
| corridor | 89.3±15.5 | 89.8±15.4 | 93.9±11.6 | 49.0±22.8 | **93.2±13.5** | 89.6±15.5 | 90.6±13.6 | **94.5±12.5** | 57.6±22.2 | **94.5±11.2** |
| *Protoss* 5_vs_5 | 38.1±24.2 | 31.4±25.2 | **54.5±25.9** | 49.0±28.2 | 54.3±24.2 | 48.4±25.9 | 41.0±24.2 | 58.8±24.5 | 54.3±24.0 | **61.5±24.8** |
| 10_vs_10 | 38.7±24.2 | 28.5±21.8 | 47.9±27.2 | 40.6±23.2 | **53.7±23.6** | 46.3±24.0 | 41.0±24.4 | 57.0±23.4 | 52.5±22.1 | **61.1±24.8** |
| 10_vs_11 | 12.7±17.4 | 12.5±16.5 | 22.3±21.0 | 18.6±18.8 | **30.7±19.8** | 22.7±22.2 | 15.6±15.9 | 27.3±24.7 | 20.9±20.9 | **34.4±24.8** |
| 20_vs_20 | 39.8±24.9 | 35.4±21.5 | 57.0±24.8 | 38.7±23.1 | **59.8±23.2** | 48.4±25.3 | 43.6±23.6 | 61.5±22.1 | 51.8±25.0 | **64.5±23.5** |
| 20_vs_23 | 15.2±18.5 | 9.0±14.2 | 22.7±21.7 | 11.1±14.6 | **23.4±19.2** | 18.0±17.4 | 9.4±14.7 | 23.4±21.4 | 12.1±15.9 | **26.4±20.8** |
| *Terran* 5_vs_5 | 27.5±24.0 | 26.2±19.5 | 36.3±24.8 | 34.2±23.4 | **39.5±24.7** | 31.1±22.9 | 34.8±23.0 | 41.0±23.7 | 36.7±24.8 | **43.0±23.0** |
| 10_vs_10 | 23.8±20.5 | 21.1±20.8 | 25.8±19.7 | 23.2±19.6 | **28.3±20.6** | 25.8±20.9 | 24.2±21.6 | 32.0±24.4 | 28.9±24.7 | **33.2±23.4** |
| 10_vs_11 | 10.2±15.4 | 7.2±13.3 | **18.2±19.4** | 11.3±15.3 | 18.2±18.7 | 11.7±17.4 | 10.4±15.2 | 17.8±17.7 | 16.4±17.8 | **21.3±20.3** |
| 20_vs_20 | 13.1±17.1 | 11.9±18.2 | 21.5±20.4 | 8.8±13.5 | **23.0±22.4** | 14.5±17.3 | 13.7±17.4 | 21.1±20.4 | 17.2±16.8 | **24.4±23.1** |
| 20_vs_23 | 3.9±10.6 | 4.1±10.3 | 5.7±11.4 | 2.3±7.3 | **7.2±12.9** | 6.4±12.2 | 3.5±9.2 | 7.2±12.6 | 4.7±10.2 | **8.6±14.8** |
| *Zerg* 5_vs_5 | 23.4±21.1 | 23.6±21.0 | 31.1±20.4 | 33.0±22.5 | **35.2±25.7** | 31.1±22.3 | 26.0±22.2 | 34.8±23.6 | 35.0±23.2 | **40.8±21.6** |
| 10_vs_10 | 25.8±21.6 | 25.8±22.5 | 32.2±24.6 | 30.7±24.0 | **34.8±22.1** | 31.4±21.9 | 31.1±24.8 | 35.5±23.9 | 33.0±25.0 | **37.9±24.0** |
| 10_vs_11 | 19.3±20.1 | 12.9±17.4 | 22.5±20.5 | 19.3±18.0 | **23.4±21.1** | 20.1±18.2 | 18.6±20.6 | 22.7±18.3 | 23.0±21.1 | **26.0±23.0** |
| 20_vs_20 | 19.9±21.0 | 11.1±16.2 | 22.5±21.4 | 5.7±10.9 | **24.8±20.8** | 22.9±21.7 | 16.0±17.3 | 27.3±22.0 | 16.4±18.1 | **31.1±24.6** |
| 20_vs_23 | 13.1±17.7 | 7.8±12.8 | 12.5±15.3 | 7.6±13.1 | **18.8±18.5** | 15.8±18.5 | 10.4±15.2 | **16.4±19.9** | 13.7±17.4 | 16.0±19.4 |

Table 1: Win rate comparison (in percentage) for SMACv1 (first 4 tasks) & SMACv2.

| Methods | Hopper-v2 | Ant-v2 | HalfCheetah-v2 |
|---|---|---|---|
| BC | 808.1 ± 39.1 | 1303.9 ± 122.0 | 4119.9 ± 350.7 |
| IIPL | 782.0 ± 81.5 | 1312.0 ± 155.6 | 4028.8 ± 430.0 |
| IPL-VDN | 846.6 ± 65.4 | 1376.1 ± 142.0 | 4287.5 ± 273.1 |
| SL-MARL | 890.0 ± 88.7 | 1334.1 ± 150.9 | 4233.9 ± 303.1 |
| O-MAPL | **1114.4 ± 154.1** | **1406.4 ± 163.7** | **4382.0 ± 189.7** |

Table 2: Return comparisons on MaMujoco tasks

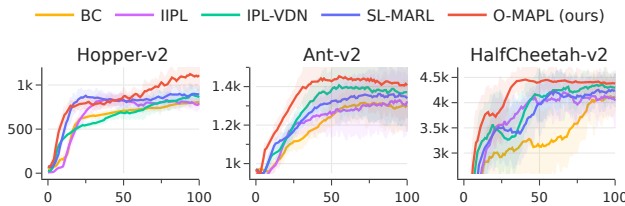

Figure 2: Evaluation curves (in returns) on MaMujoco tasks

**Dataset.** There are no human-labeled preference datasets for MARL, so we create datasets for each task using two methods: (i) Rule-based method – followed by IPL (Hejna & Sadigh, 2024), we sample trajectory pairs from offline datasets of varying quality (e.g., poor, medium, expert) and assign binary preference labels based on dataset quality; and (ii) LLM-based method – followed by DPM (Kang et al., 2024), we sample pairs from offline datasets and use GPT-4o to annotate labels with prompts constructed from the global state of each trajectory (details in the appendix).

Specifically, we used offline datasets of varying quality from OMIGA (Wang et al., 2022) and ComaDICE (Bui et al., 2025), sampling one thousand pairs for MaMujoco tasks and two thousand pairs for SMAC tasks. For MaMujoco, we selected "medium-replay", "medium", and "expert" instances, while for SMACv1, we chose "poor", "medium", and "good" instances. Note that ComaDICE only provides a "medium" dataset for SMACv2, therefore, we generated new "poor" and "expert" datasets for SMACv2. Additionally, LLM-based prompts require detailed information from trajectory states (e.g., SMAC: remaining health points, shields, relative positions, cooldown time, agent types, action meanings), which we cannot extract from MaMujoco states. Therefore, we have no LLM-based dataset for MaMujoco tasks.

**Baselines.** We consider the following baselines for our evaluations: (i) **Behavioral Cloning (BC)** trains a policy by directly imitating all prefered trajectories in the dataset $\mathcal{P}$; (ii) **Independent IPL (IIPL)** is a straightforward extension of the IPL approach (Hejna & Sadigh, 2024) to multi-agent learning, where the single-agent IPL algorithm is applied independently to each agent; (iii) **Supervised Learning MARL (SL-MARL)** is a two-phase approach where we first learn the reward function and then use it to train a policy with OMIGA (Wang et al., 2022), a state-of-the-art MARL algorithm, serving as the offline counterpart of the two-phase approach in (Kang et al., 2024); and (iv) **IPL-VDN**, which is similar to our algorithm but without the mixing networks, instead employing the standard VDN approach (Sunehag et al., 2017) to aggregate local Q and V functions

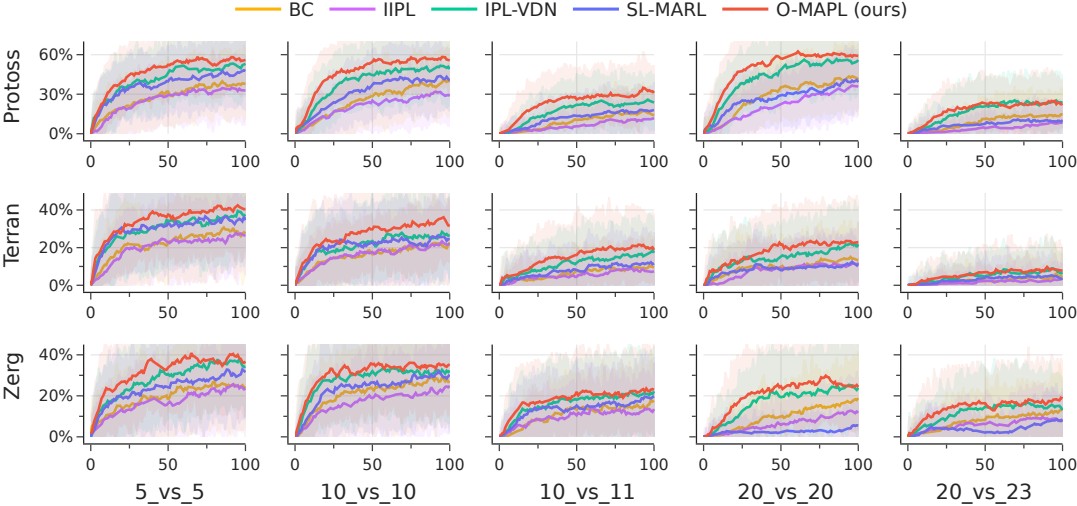

Figure 1: Evaluation curves (in win rates) of our O-MAPL for SMACv2 with rule-based preference data.

via a simple linear combination with unit weights.

**Results.** Overall, our experimental results demonstrate the effectiveness of O-MAPL in both continuous and discrete multi-agent reinforcement learning environments. In the following, we highlight some of our main results. Due to limited space, all remaining results are in our appendix.

Table 1 and 2 provide a detailed comparison of win rates for SMACv1 and SMACv2 tasks and of returns for MaMujoco. O-MAPL achieves the highest win rates/returns across most tasks, outperforming all baseline methods. For example, in the *2c_vs_64zg* task, O-MAPL achieves a win rate of 74.4%, significantly surpassing other methods. In *corridor*, O-MAPL achieves a win rate of 93.2%, showcasing its ability to handle structured navigation tasks effectively.

Furthermore, Table 1 demonstrates that our algorithm, O-MAPL, achieves higher win rates in most SMAC tasks when using LLM-generated data than when using the ruled-based generated data. This finding highlights the potential of leveraging LLMs for rich and cost-effective data generation, substantially improving environment understanding and policy learning in complex multi-agent tasks.

Finally, we present evaluation curves for both SMACv2 (Figure 1) and MaMujoco tasks (Figure 2). The results show that O-MAPL consistently and significantly outperforms other baselines throughout the training process. Our algorithm converges faster, achieving high win rates and returns at earlier training stages across most tasks. This demonstrates the effectiveness of our multi-agent end-to-end preference learning approach, supported by a systematic and carefully designed value decomposition.

Additional details on dataset generation, hype-parameters,

and detailed returns and win rates for all tasks can be found in the appendix.

## 7. Conclusion

**Summary.** We explored preference-based learning in multi-agent environments, proposing a novel end-to-end method based on the MaxEnt RL framework that eliminates the need for explicit reward modeling. To facilitate efficient training, we developed a new value factorization approach that learns the global preference-based loss function by updating local value functions. Key properties, including global-local consistency and convexity, were thoroughly examined. Extensive experiments on both rule-based and LLM-based datasets show that our algorithm outperforms existing methods across multiple benchmark tasks in the MAMuJoCo and SMAC environments.

**Limitations and Future Work.** The strong performance of LLM-based preference data suggests that leveraging LLMs, coupled with a systematic value factorization approach, can be highly effective for training policies in complex multi-agent environments. This opens promising avenues for using LLMs to enhance both environment understanding and policy learning. However, our work has some limitations that need further exploration. For example, we primarily focus on cooperative learning, while more challenging mixed cooperative-competitive environments would require different methodologies. Additionally, our method still depends on a large number of preference-based demonstrations for optimal policy learning. Although LLMs can quickly generate extensive demonstrations, improving sample efficiency remains a key challenge, particularly when data must be collected from real human feedback.

## Acknowledgment

This work is supported by the Lee Kong Chian Fellowship awarded to Tien Mai.

## Impact Statement

This paper presents work whose goal is to advance the field of Machine Learning, in particular multi-agent reinforcement learning. There are many potential societal consequences of our work, none which we feel must be specifically highlighted here.

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

# A. Missing Proofs

We provide proofs that are omitted in the main paper.

## A.1. Proof of Proposition 4.1

**Proposition 4.1:**    *The preference-based loss function $\mathcal{L}(\mathbf{q}, \mathbf{v}, w)$ is concave in $\mathbf{q}$ and $w$ (the parameters of the mixing networks), while the extreme-V loss function $\mathcal{J}(\mathbf{v})$ is convex in $\mathbf{v}$.*

*Proof.* We first recall that the preference-based loss function has the following form:

$$\mathcal{L}(\mathbf{q}, \mathbf{v}, w) = \sum_{(\sigma_1, \sigma_2) \in \mathcal{P}} \sum_{(\mathbf{s}, \mathbf{a}) \in \sigma_1} R_w[\mathbf{q}, \mathbf{v}](\mathbf{s}, \mathbf{a}) - \log\left(e^{\sum_{\sigma_1} R_w[\mathbf{q}, \mathbf{v}](\mathbf{s}, \mathbf{a})} + e^{\sum_{\sigma_2} R_w[\mathbf{q}, \mathbf{v}](\mathbf{s}, \mathbf{a})}\right) + \phi(R_w[\mathbf{q}, \mathbf{v}](\mathbf{s}, \mathbf{a})).$$

We observe that under the assumption that the mixing networks are linear in their inputs, the function $R_w[\mathbf{q}, \mathbf{v}](\mathbf{s}, \mathbf{a})$ is linear in $\mathbf{q}(\mathbf{s}, \mathbf{a})$ and $\theta$. This implies that for any $\alpha \in [0, 1]$ and for any two vectors of local Q values $\mathbf{q}^1, \mathbf{q}^2$, we have:

$$\alpha R_w[\mathbf{q}^1, \mathbf{v}](\mathbf{s}, \mathbf{a}) + (1 - \alpha) R_w[\mathbf{q}^2, \mathbf{v}](\mathbf{s}, \mathbf{a}) = R_w[\alpha \mathbf{q}^1 + (1 - \alpha)\mathbf{q}^2, \mathbf{v}](\mathbf{s}, \mathbf{a}).$$

Now consider the term $\phi(R_w[\mathbf{q}, \mathbf{v}](\mathbf{s}, \mathbf{a}))$. Since $\phi$ is concave, we have the following inequality for any $\alpha \in (0, 1)$ and two vectors $\mathbf{q}^1, \mathbf{q}^2$:

$$\alpha \phi(R_w[\mathbf{q}^1, \mathbf{v}](\mathbf{s}, \mathbf{a})) + (1 - \alpha)\phi(R_w[\mathbf{q}^2, \mathbf{v}](\mathbf{s}, \mathbf{a})) \leq \phi\left(\alpha R_w[\mathbf{q}^1, \mathbf{v}](\mathbf{s}, \mathbf{a}) + (1 - \alpha) R_w[\mathbf{q}^2, \mathbf{v}](\mathbf{s}, \mathbf{a})\right)$$
$$\leq \phi\left(R_w[\alpha \mathbf{q}^1 + (1 - \alpha)\mathbf{q}^2, \mathbf{v}](\mathbf{s}, \mathbf{a})\right), \tag{8}$$

which implies the concavity of $\phi(R_w[\mathbf{q}, \mathbf{v}](\mathbf{s}, \mathbf{a}))$ in $\mathbf{q}$.

For the term $\log\left(e^{\sum_{\sigma_1} R_w[\mathbf{q}, \mathbf{v}](\mathbf{s}, \mathbf{a})} + e^{\sum_{\sigma_2} R_w[\mathbf{q}, \mathbf{v}](\mathbf{s}, \mathbf{a})}\right)$, we note the following. First:

$$\alpha \sum_{\sigma} R_w[\mathbf{q}^1, \mathbf{v}](\mathbf{s}, \mathbf{a}) + (1 - \alpha) \sum_{\sigma} R_w[\mathbf{q}^2, \mathbf{v}](\mathbf{s}, \mathbf{a}) = \sum_{\sigma} R_w[\alpha \mathbf{q}^1 + (1 - \alpha)\mathbf{q}^2, \mathbf{v}](\mathbf{s}, \mathbf{a}),$$

for any trajectory $\sigma$. Moreover, since the log-sum-exp function $\log(e^{t_1} + e^{t_2})$ is convex in $(t_1, t_2)$, we also have the following inequalities for any $\alpha \in (0, 1)$ and two vectors $\mathbf{q}^1, \mathbf{q}^2$:

$$\alpha \log\left(e^{\sum_{\sigma_1} R_w[\mathbf{q}^1, \mathbf{v}](\mathbf{s}, \mathbf{a})} + e^{\sum_{\sigma_2} R_w[\mathbf{q}^1, \mathbf{v}](\mathbf{s}, \mathbf{a})}\right) + (1 - \alpha) \log\left(e^{\sum_{\sigma_1} R_w[\mathbf{q}^2, \mathbf{v}](\mathbf{s}, \mathbf{a})} + e^{\sum_{\sigma_2} R_w[\mathbf{q}^2, \mathbf{v}](\mathbf{s}, \mathbf{a})}\right)$$
$$\leq \log\left(e^{\alpha \sum_{\sigma_1} R_w[\mathbf{q}^1, \mathbf{v}](\mathbf{s}, \mathbf{a}) + (1 - \alpha) \sum_{\sigma_1} R_w[\mathbf{q}^2, \mathbf{v}](\mathbf{s}, \mathbf{a})} + e^{\alpha \sum_{\sigma_2} R_w[\mathbf{q}^1, \mathbf{v}](\mathbf{s}, \mathbf{a}) + (1 - \alpha) \sum_{\sigma_2} R_w[\mathbf{q}^2, \mathbf{v}](\mathbf{s}, \mathbf{a})}\right)$$
$$= \log\left(e^{R_w[\alpha \mathbf{q}^1 + (1 - \alpha)\mathbf{q}^2, \mathbf{v}](\mathbf{s}, \mathbf{a})} + e^{R_w[\alpha \mathbf{q}^1 + (1 - \alpha)\mathbf{q}^2, \mathbf{v}](\mathbf{s}, \mathbf{a})}\right),$$

which implies that $\log\left(e^{\sum_{\sigma_1} R_w[\mathbf{q}, \mathbf{v}](\mathbf{s}, \mathbf{a})} + e^{\sum_{\sigma_2} R_w[\mathbf{q}, \mathbf{v}](\mathbf{s}, \mathbf{a})}\right)$ is convex in $\mathbf{q}$.

Putting all the above together, we see that $\mathcal{L}(\mathbf{q}, \mathbf{v}, w)$ is concave in $\mathbf{q}$.

Finally, since the mixing networks are linear in $\mathbf{q}$ and $w$, a similar argument shows that $\mathcal{L}(\mathbf{q}, \mathbf{v}, w)$ is also concave in $w$.

For the convexity of the extreme-V function $\mathcal{J}(\mathbf{v})$, we rewrite the function as:

$$\mathcal{J}(\mathbf{v}) = \mathbb{E}_{(\mathbf{s}, \mathbf{a}) \sim \mu_{\text{tot}}} \left[e^{\frac{\mathcal{M}_w[\mathbf{q}(\mathbf{s}, \mathbf{a})] - \mathcal{M}_w[\mathbf{v}(\mathbf{s})]}{\beta}}\right] - \mathbb{E}_{(\mathbf{s}, \mathbf{a}) \sim \mu_{\text{tot}}} \left[\frac{\mathcal{M}_w[\mathbf{q}(\mathbf{s}, \mathbf{a})] - \mathcal{M}_w[\mathbf{v}(\mathbf{s})]}{\beta}\right] - 1.$$

Since the mixing network $\mathcal{M}_w[\mathbf{v}]$ is linear in $\mathbf{v}$, we can see that the term

$$\mathbb{E}_{(\mathbf{s}, \mathbf{a}) \sim \mu_{\text{tot}}} \left[\frac{\mathcal{M}_w[\mathbf{q}(\mathbf{s}, \mathbf{a})] - \mathcal{M}_w[\mathbf{v}(\mathbf{s})]}{\beta}\right]$$

is also linear in $\mathbf{v}$.

Moreover, the exponential function $e^x$ is always convex in $x$. Thus, in a similar way as shown above, we can prove that

$e^{\frac{\mathcal{M}_w[\mathbf{q}(\mathbf{s}, \mathbf{a})] - \mathcal{M}_w[\mathbf{v}(\mathbf{s})]}{\beta}}$ is convex in $\mathbf{v}$. All these observations imply that $\mathcal{J}(\mathbf{v})$ is convex in $\mathbf{v}$, as desired. $\qquad\square$

## A.2. Proof of Proposition 4.2

**Proposition 4.1:** *If the mixing networks $\mathcal{M}_w[\mathbf{q}]$ and $\mathcal{M}_w[\mathbf{v}]$ are two-layer (or multi-layer) feed-forward networks, the preference-based loss function $\mathcal{L}(\mathbf{q}, \mathbf{v}, w)$ is no longer concave in $\mathbf{q}$ or $w$, and the extreme-V loss function $\mathcal{J}(\mathbf{v})$ is **not** convex in $\mathbf{v}$.*

*Proof.* Following standard settings in value factorization, a 2-layer mixing network is typically constructed with non-negative weights and convex activations (e.g., ReLU). Under this setting, according to (Bui et al., 2024), $\mathcal{M}_w[\mathbf{q}]$ and $\mathcal{M}_w[\mathbf{v}]$ are convex in $\mathbf{q}$ and $\mathbf{v}$, respectively.

From this observation, we first recall the preference-based loss function:

$$\mathcal{L}(\mathbf{q}, \mathbf{v}, w) = \sum_{(\sigma_1, \sigma_2) \in \mathcal{P}} \sum_{(\mathbf{s}, \mathbf{a}) \in \sigma_1} R_w[\mathbf{q}, \mathbf{v}](\mathbf{s}, \mathbf{a}) - \log\left(e^{\sum_{\sigma_1} R_w[\mathbf{q}, \mathbf{v}](\mathbf{s}, \mathbf{a})} + e^{\sum_{\sigma_2} R_w[\mathbf{q}, \mathbf{v}](\mathbf{s}, \mathbf{a})}\right) + \phi(R_w[\mathbf{q}, \mathbf{v}](\mathbf{s}, \mathbf{a})).$$

It can be seen that the first term of $\mathcal{L}(\mathbf{q}, \mathbf{v}, w)$ involves $R_w[\mathbf{q}, \mathbf{v}](\mathbf{s}, \mathbf{a})$, which can be written as:

$$R_w[\mathbf{q}, \mathbf{v}](\mathbf{s}, \mathbf{a}) = \mathcal{M}_w[\mathbf{q}(\mathbf{s}, \mathbf{a})] - \gamma \mathbb{E}_{\mathbf{s}'}\left[\mathcal{M}_w[\mathbf{v}(\mathbf{s}')]\right].$$

Since $\mathcal{M}_w[\mathbf{q}(\mathbf{s}, \mathbf{a})]$ is convex in $\mathbf{q}$, the first term of $\mathcal{L}(\mathbf{q}, \mathbf{v}, w)$ is convex in $\mathbf{q}$, which generally implies that this function is not concave in $\mathbf{q}$.

In a similar way, since the the mixing function $\mathcal{M}_w[\mathbf{q}(\mathbf{s}, \mathbf{a})]$ is also convex in $w$, implying that $\mathcal{L}(\mathbf{q}, \mathbf{v}, w)$ is also not concave in $w$.

To prove the non-convexity of the Extreme-V function $J(\mathbf{v})$, we recall that:

$$\mathcal{J}(\mathbf{v}) = \mathbb{E}_{(\mathbf{s}, \mathbf{a}) \sim \mu_{\text{tot}}}\left[e^{\frac{\mathcal{M}_w[\mathbf{q}(\mathbf{s}, \mathbf{a})] - \mathcal{M}_w[\mathbf{v}(\mathbf{s})]}{\beta}}\right] - \mathbb{E}_{(\mathbf{s}, \mathbf{a}) \sim \mu_{\text{tot}}}\left[\frac{\mathcal{M}_w[\mathbf{q}(\mathbf{s}, \mathbf{a})] - \mathcal{M}_w[\mathbf{v}(\mathbf{s})]}{\beta}\right] - 1.$$

We will find a counterexample to show that $J(\mathbf{v})$ is not convex under a 2-layer mixing network. For simplicity, since $\mathcal{M}_w[\mathbf{q}]$ is fixed in $J(\mathbf{v})$, we select $\mathcal{M}_w[\mathbf{q}](\mathbf{s}, \mathbf{a}) = 0$. We then create a simple example where there is only one agent (i.e., $\mathbf{v}(\mathbf{s}) = \{v_1(s_1)\}$), and the mixing network $\mathcal{M}_w[\mathbf{v}]$ takes a one-dimensional input with a ReLU activation (a commonly used activation function in the context). Specifically, we can write $\mathcal{M}_w[\mathbf{v}]$ as:

$$\mathcal{M}_w[\mathbf{v}(\mathbf{s})] = \begin{cases} v_1(s_1) & \text{if } v_1(s_1) > 0, \\ e^{v_1(s_1)} - 1 & \text{if } v_1(s_1) \le 0. \end{cases}$$

Then, for a given pair $(\mathbf{s}, \mathbf{a})$, the corresponding term in $J(\mathbf{v})$ associated with $(\mathbf{s}, \mathbf{a})$ can be written as:

$$e^{1 - e^{v_1}} + (e^{v_1} - 1).$$

Here, for simplicity, we select $\beta = 1$, omit the notation $s_1$ in the function $v_1(s_1)$, and only consider the case where $v_1 \le 0$. We see that the function $f(t) = e^{1 - e^t} + (e^t - 1)$ is not convex for $t \le 0$ (see the plot of this function in Figure 3).

$\square$

## A.3. Proof of Theorem 4.3

**Theorem 4.3:** *Let $\pi_i^*$ be the optimal solution to the local WBC problem in (4). Then, the global policy $\pi_{tot}^*$, defined as $\pi_{tot}^*(\mathbf{s}, \mathbf{a}) = \prod_i \pi_i^*(a_i|s_i)$, is also optimal for the global WBC problem in (3). In other words, the local WBC approach yields local policies that are consistent with the desired globally optimal policy.*

*Proof.* For notational simplicity, let $G(\pi_{\text{tot}})$ be the objective function of the global WBC problem:

$$G(\pi_{\text{tot}}) = \mathbb{E}_{\mathbf{s}, \mathbf{a} \sim \mu_{\text{tot}}}\left[e^{\frac{Q_{\text{tot}}(\mathbf{s}, \mathbf{a}) - V_{\text{tot}}(\mathbf{s})}{\beta}} \log \pi_{\text{tot}}(\mathbf{a}|\mathbf{s})\right].$$

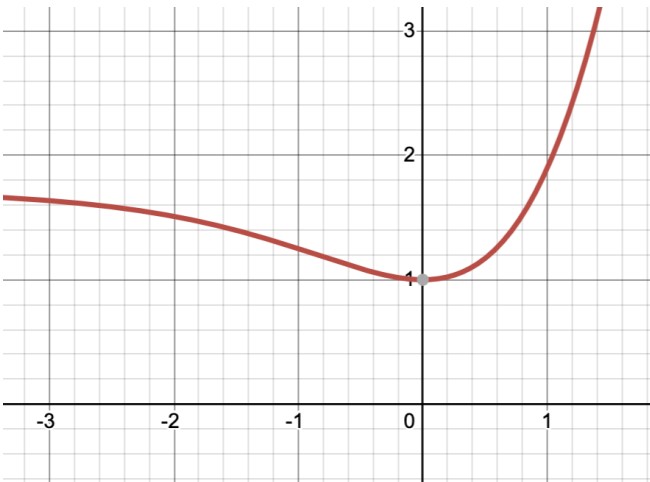

Figure 3: Plot of the function $f(t) = e^{1-e^t} + e^t - 1$.

Since we are seeking a decomposable policy $\pi_{\text{tot}}(\mathbf{a}|\mathbf{s}) = \prod_{i \in \mathcal{N}} \pi_i(a_i|s_i)$, we have, for any $\pi_{\text{tot}} \in \Pi_{\text{tot}}$ such that $\pi_{\text{tot}} = \prod_i \pi_i$:

$$
\begin{aligned}
G(\pi_{\text{tot}}) &= \mathbb{E}_{\mathbf{s},\mathbf{a} \sim \mu_{\text{tot}}} \left[ e^{\frac{Q_{\text{tot}}(\mathbf{s},\mathbf{a}) - V_{\text{tot}}(\mathbf{s})}{\beta}} \log \prod_i \pi_i(a_i|s_i) \right] \\
&= \sum_{i \in \mathcal{N}} \left\{ \mathbb{E}_{\mathbf{s},\mathbf{a} \sim \mu_{\text{tot}}} \left[ e^{\frac{Q_{\text{tot}}(\mathbf{s},\mathbf{a}) - V_{\text{tot}}(\mathbf{s})}{\beta}} \log \pi_i(a_i|s_i) \right] \right\} \\
&\stackrel{(a)}{\leq} \sum_{i \in \mathcal{N}} \left\{ \mathbb{E}_{\mathbf{s},\mathbf{a} \sim \mu_{\text{tot}}} \left[ e^{\frac{Q_{\text{tot}}(\mathbf{s},\mathbf{a}) - V_{\text{tot}}(\mathbf{s})}{\beta}} \log \pi_i^*(a_i|s_i) \right] \right\} \\
&= \mathbb{E}_{\mathbf{s},\mathbf{a} \sim \mu_{\text{tot}}} \left[ e^{\frac{Q_{\text{tot}}(\mathbf{s},\mathbf{a}) - V_{\text{tot}}(\mathbf{s})}{\beta}} \log \pi_{\text{tot}}^*(\mathbf{a}|\mathbf{s}) \right],
\end{aligned}
$$

where $(a)$ holds because each $\pi_i^*$ is optimal for the corresponding local WBC problem. Thus, we have $G(\pi_{\text{tot}}) \leq G(\pi_{\text{tot}}^*)$ for any $\pi_{\text{tot}} \in \Pi_{\text{tot}}$, implying that $\pi_{\text{tot}}^*$ is also optimal for the global WBC. This establishes the GLC, as desired. $\square$

### A.4. Proof of Theorem 4.4

**Theorem 4.4:** *Let $\pi_i^*$ be optimal to the local WBC, then the following equality holds for all $s_i \in \mathcal{S}_i, a_i \in \mathcal{A}_i$:*

$$
\pi_i^*(a_i|s_i) = \frac{\eta(s_i)}{\Delta(s_i)} \mu_i(a_i|s_i) e^{\frac{w_i^q q_i(s_i,a_i) - w_i^v v_i(s_i)}{\beta}} \tag{9}
$$

*where $w_i^q$ and $w_i^v$ are parameters of the mixing networks $\mathcal{M}_w[\boldsymbol{q}]$, $\mathcal{M}_w[\boldsymbol{v}]$, respectively. In addition, $\eta(s_i)/\Delta(s_i)$ is a correction term defined as follows:*

$$
\begin{aligned}
\eta(s_i) &= \sum_{\mathbf{s}',\mathbf{a}'|s_i'=s_i} e^{\frac{b_q - b_v}{\beta}} \prod_{j \in \mathcal{N}, j \neq i} \mu(a_j'|s_j') e^{\frac{w_j^q q_j(s_j',a_j') - w_j^v v_j(s_j')}{\beta}} \\
\Delta(s_i) &= \sum_{a_i \in \mathcal{A}_i} \eta(s_i) \mu_i(a_i|s_i) e^{\frac{w_i^q q_i(s_i,a_i) - w_i^v v_i(s_i)}{\beta}}.
\end{aligned} \tag{10}
$$

*Proof.* We first note that each mixing network $\mathcal{M}_w[\mathbf{q}]$ or $\mathcal{M}_w[\mathbf{v}]$ can be expressed as a linear function of its inputs:

$$
\begin{aligned}
Q_{\text{tot}}(\mathbf{s}, \mathbf{a}) &= \mathcal{M}_w[\mathbf{q}](\mathbf{s}, \mathbf{a}) = \sum_{i \in \mathcal{N}} w_i^q q_i(s_i, a_i) + b_q, \\
V_{\text{tot}}(\mathbf{s}) &= \mathcal{M}_w[\mathbf{v}](\mathbf{s}) = \sum_{i \in \mathcal{N}} w_i^v v_i(s_i) + b_v.
\end{aligned}
$$

Thus,

$$e^{\frac{Q_{\text{tot}}(\mathbf{s},\mathbf{a}) - V_{\text{tot}}(\mathbf{s})}{\beta}} = e^{\frac{b_q - b_v}{\beta}} \prod_{i \in \mathcal{N}} e^{\frac{w_i^q q_i(s_i, a_i) - w_i^v v_i(s_i)}{\beta}}.$$

Now, let us consider the objective function of the local WBC and write:

$$g(\pi_i) = \sum_{\mathbf{s} \in \mathcal{S}, \mathbf{a} \in \mathcal{A}} \mu_{\text{tot}}(\mathbf{a}|\mathbf{s}) e^{\frac{Q_{\text{tot}}(\mathbf{s},\mathbf{a}) - V_{\text{tot}}(\mathbf{s})}{\beta}} \log \pi_i(a_i|s_i)$$

$$= \sum_{\mathbf{s} \in \mathcal{S}, \mathbf{a} \in \mathcal{A}} e^{\frac{b_q - b_v}{\beta}} \prod_{i \in \mathcal{N}} \mu(a_i|s_i) e^{\frac{w_i^q q_i(s_i, a_i) - w_i^v v_i(s_i)}{\beta}} \log \pi_i(a_i|s_i).$$

Thus, for each agent $i \in \mathcal{N}$ and local state $s_i \in \mathcal{S}_i$, we extract all the components of $g(\pi_i)$ that involve $\pi_i(a_i|s_i)$ as:

$$g^{s_i}(\pi_i) = \sum_{\mathbf{s}', \mathbf{a}'|s_i'=s_i} e^{\frac{b_q - b_v}{\beta}} \prod_{j \in \mathcal{N}, j \neq i} \mu(a_j'|s_j') e^{\frac{w_j^q q_j(s_j', a_j') - w_j^v v_j(s_j')}{\beta}}$$

$$\times \left( \mu_i(a_i'|s_i) e^{\frac{w_i^q q_i(s_i, a_i') - w_i^v v_i(s_i)}{\beta}} \log \pi_i(a_i'|s_i) \right)$$

$$= \sum_{a_i' \in \mathcal{A}_i} \eta(s_i) \left( \mu_i(a_i'|s_i) e^{\frac{w_i^q q_i(s_i, a_i') - w_i^v v_i(s_i)}{\beta}} \log \pi_i(a_i'|s_i) \right),$$

where

$$\eta(s_i) = \sum_{\mathbf{s}', \mathbf{a}'|s_i'=s_i} e^{\frac{b_q - b_v}{\beta}} \prod_{j \in \mathcal{N}, j \neq i} \mu(a_j'|s_j') e^{\frac{w_j^q q_j(s_j', a_j') - w_j^v v_j(s_j')}{\beta}},$$

which is independent of any local actions $a_i'$.

The local WBC problem thus becomes the problem of finding local policies $\pi_i(\cdot|s_i)$ that maximize $g^{s_i}(\pi_i)$ for any local state $s_i$. For notational simplicity, let

$$\delta(a_i', s_i) = \eta(s_i) \mu_i(a_i'|s_i) e^{\frac{w_i^q q_i(s_i, a_i') - w_i^v v_i(s_i)}{\beta}}.$$

We then write the local objective function $g^{s_i}(\pi_i)$ as:

$$g^{s_i}(\pi_i) = \sum_{a_i \in \mathcal{A}_i} \delta(a_i, s_i) \log \pi_i(a_i|s_i).$$

To solve the problem $\max_{\pi_i} g^{s_i}(\pi_i)$, let us consider a general version (with simplified notation):

$$\max_{\mathbf{t}} \left\{ g(\mathbf{t}) = \sum_{i \in \mathcal{N}} \alpha_i \log t_i \;\middle|\; \mathbf{t} \in [0,1]^n, \sum_i t_i = 1 \right\},$$

where $\alpha_i \geq 0$ and $g(\mathbf{t}) : [0,1]^n \to \mathbb{R}$. By considering the Lagrangian dual of this problem, we can see that an optimal solution $\mathbf{t}^*$ must satisfy the following KKT conditions:

$$\begin{cases} \mathbf{t}^* \in (0,1)^n, \\ \sum_i t_i^* = 1, \\ \frac{\alpha_i}{t_i^*} = \frac{\alpha_j}{t_j^*}, \quad \forall i, j \in \mathcal{N}. \end{cases}$$

These conditions directly imply that:

$$t_i^* = \frac{\alpha_i}{\sum_{j \in \mathcal{N}} \alpha_j}.$$

We return to the maximization of $g^{s_i}(\pi_i)$, which yields an optimal solution:

$$\pi_i^*(a_i|s_i) = \frac{\delta(a_i, s_i)}{\sum_{a_i' \in \mathcal{A}_i} \delta(a_i', s_i)}, \quad \forall a_i \in \mathcal{A}_i.$$

Putting everything together, we see that the following solution $\pi_i^*$ is optimal for the local WBC:

$$\pi_i(a_i|s_i) = \frac{\eta(s_i)\mu_i(a_i|s_i)e^{\frac{w_i^q q_i(s_i, a_i) - w_i^v v_i(s_i)}{\beta}}}{\Delta(s_i)},$$

where

$$\Delta(s_i) = \sum_{a_i \in \mathcal{A}_i} \eta(s_i)\mu_i(a_i|s_i)e^{\frac{w_i^q q_i(s_i, a_i) - w_i^v v_i(s_i)}{\beta}}.$$

$\square$

### A.5. Proof of Proposition 4.5

**Proposition 4.5:** *Each local value $v_i$ can be expressed as a (modified) log-sum-exp of the local Q-function $q_i$:*

$$v_i(s_i) = \frac{\beta}{w_i^v} \log \sum_{a_i \sim \mu_i(\cdot|s_i)} e^{\frac{w_i^q}{\beta} q_i(s_i, a_i)} + \frac{\beta}{w_i^v} \log\left(\frac{\eta(s_i)}{\Delta(s_i)}\right). \tag{11}$$

*Proof.* Since $\pi_i^*$ is a valid probability distribution, we have $\sum_{a_i} \pi_i^*(a_i|s_i) = 1$. Substituting the closed-form formula of $\pi_i^*$ stated in Theorem 4.4, we have:

$$\sum_{a_i} \frac{\eta(s_i)}{\Delta(s_i)}\mu_i(a_i|s_i)e^{\frac{w_i^q q_i(s_i, a_i) - w_i^v v_i(s_i)}{\beta}} = 1.$$

Taking $e^{w_i^v v_i(s_i)/\beta}$ outside the summation, we get:

$$\sum_{a_i} \frac{\eta(s_i)}{\Delta(s_i)}\mu_i(a_i|s_i)e^{\frac{w_i^q q_i(s_i, a_i)}{\beta}} = e^{w_i^v v_i(s_i)/\beta}.$$

This directly leads to the log-sum-exp formula:

$$v_i(s_i) = \frac{\beta}{w_i^v} \log\left(\sum_{a_i} \frac{\eta(s_i)}{\Delta(s_i)}\mu_i(a_i|s_i)e^{\frac{w_i^q q_i(s_i, a_i)}{\beta}}\right) = \frac{\beta}{w_i^v} \log \sum_{a_i \sim \mu_i(\cdot|s_i)} e^{\frac{w_i^q}{\beta} q_i(s_i, a_i)} + \frac{\beta}{w_i^v} \log\left(\frac{\eta(s_i)}{\Delta(s_i)}\right).,$$

as desired.

$\square$

## B. Additional Details

### B.1. Offline Preference Multi-Agent Datasets

In this section, we provide a detailed description of how we constructed the dataset for preference learning tasks. Our datasets span both discrete and continuous domains, covering the environments SMACv1, SMACv2, and MaMujoco. The datasets are designed to include varying qualities of data, sampled trajectory pairs, and their preference labels to facilitate preference learning. To create datasets suitable for preference learning, we sampled trajectory pairs from varying quality offline datasets and generated preference labels. The labeling process was performed using two approaches:

- **Rule-based Methods:** Following IPL (Hejna & Sadigh, 2024), we sampled trajectory pairs and assigned binary preference labels based on dataset quality (e.g., poor, medium, expert).

- **LLM-based Methods:** Following DPM (Kang et al., 2024), we sampled trajectory pairs and annotated them using preference policies from large language models (e.g., Llama 3, GPT-4o).

For MaMujoco tasks, 1k trajectory pairs were sampled, while for SMAC tasks, 2k trajectory pairs were sampled. Table 3 summarizes the dataset details, including state dimensions, action dimensions, sample sizes, and average returns.

The datasets constructed for this study span a diverse range of environments and tasks, ensuring comprehensive evaluation of preference learning algorithms. The inclusion of varying quality levels and both rule-based and LLM-based labeling methods provides a robust foundation for preference-based multi-agent reinforcement learning research.

| | Tasks | State dim | Obs. dim | Act. dim | Samples | Max len. | Avg. returns | File size |
|---|---|---|---|---|---|---|---|---|
| Ma-Mujoco | Hopper-v2 | 42 | 14 | 1 | 1000 | 1000 | 1354.0±1121.6 | 255 MB |
| | Ant-v2 | 226 | 113 | 4 | 1000 | 1000 | 1514.9±435.8 | 1003 MB |
| | HalfCheetah-v2 | 138 | 23 | 1 | 1000 | 1000 | 1640.5±1175.7 | 1802 MB |
| SMACv1 | 2c_vs_64zg | 1350 | 478 | 70 | 2000 | 280 | 13.99±4.75 | 401 MB |
| | 5m_vs_6m | 780 | 124 | 12 | 2000 | 36 | 13.26±5.02 | 72 MB |
| | 6h_vs_8z | 1278 | 172 | 14 | 2000 | 48 | 13.01±3.95 | 182 MB |
| | corridor | 2610 | 346 | 30 | 2000 | 394 | 12.69±6.30 | 979 MB |
| SMACv2 | protoss_5_vs_5 | 130 | 92 | 11 | 2000 | 142 | 16.07±4.94 | 56 MB |
| | protoss_10_vs_10 | 310 | 182 | 16 | 2000 | 178 | 15.72±4.28 | 209 MB |
| | protoss_10_vs_11 | 327 | 191 | 17 | 2000 | 146 | 15.45±4.85 | 218 MB |
| | protoss_20_vs_20 | 820 | 362 | 26 | 2000 | 200 | 15.63±4.76 | 726 MB |
| | protoss_20_vs_23 | 901 | 389 | 29 | 2000 | 200 | 14.44±4.73 | 799 MB |
| | terran_5_vs_5 | 120 | 82 | 11 | 2000 | 200 | 16.20±6.37 | 44 MB |
| | terran_10_vs_10 | 290 | 162 | 16 | 2000 | 200 | 14.86±5.78 | 151 MB |
| | terran_10_vs_11 | 306 | 170 | 17 | 2000 | 200 | 13.52±5.44 | 165 MB |
| | terran_20_vs_20 | 780 | 322 | 26 | 2000 | 200 | 13.52±5.76 | 530 MB |
| | terran_20_vs_23 | 858 | 346 | 29 | 2000 | 200 | 10.67±5.11 | 563 MB |
| | zerg_5_vs_5 | 120 | 82 | 11 | 2000 | 57 | 14.79±7.70 | 31 MB |
| | zerg_10_vs_10 | 290 | 162 | 16 | 2000 | 70 | 14.61±5.63 | 99 MB |
| | zerg_10_vs_11 | 306 | 170 | 17 | 2000 | 104 | 13.67±5.71 | 101 MB |
| | zerg_20_vs_20 | 780 | 322 | 26 | 2000 | 134 | 12.14±3.95 | 303 MB |
| | zerg_20_vs_23 | 858 | 346 | 29 | 2000 | 99 | 10.88±4.36 | 313 MB |

Table 3: Datasets

### B.1.1. SMAC DATASET

SMACv1 (Samvelyan et al., 2019) is a benchmark environment for cooperative multi-agent reinforcement learning (MARL), built on Blizzard's StarCraft II RTS game. It leverages the StarCraft II Machine Learning API and DeepMind's PySC2 to enable autonomous agent interaction with StarCraft II. Unlike PySC2, SMACv1 focuses on decentralized micromanagement scenarios, where each unit is controlled by an individual RL agent.

We evaluate on the following tasks: 2c_vs_64zg, 5m_vs_6m, 6h_vs_8z, and corridor. Among these, 2c_vs_64zg and 5m_vs_6m are categorized as hard tasks, while 6h_vs_8z and corridor are considered super hard. The offline dataset for SMACv1 was sourced from the work of Meng et al., where MAPPO was used to train agents. These agents were then used to generate offline datasets for the community. The dataset quality varies across poor, medium, and good levels, ensuring comprehensive coverage of different learning stages.

SMACv2 (Ellis et al., 2022) builds upon SMACv1, introducing enhancements to challenge contemporary MARL algorithms. It incorporates randomized start positions, randomized unit types, and adjustments to unit sight and attack ranges. These changes increase the diversity of agent interactions and align the sight range with the true values in StarCraft II. Tasks in SMACv2 are grouped by factions (protoss, terran, zerg) and instances (5_vs_5, 10_vs_10, 10_vs_11, 20_vs_20, 20_vs_23). The difficulty increases progressively from 5_vs_5 to 20_vs_23.

The offline dataset for SMACv2 was derived from the ComaDICE paper (Bui et al., 2025), where MAPPO (Yu et al., 2022) was used to train agents over 10e6 steps, followed by random sampling of 1k trajectories. This dataset primarily represents medium-quality data. To ensure varying quality levels, we created additional datasets for poor and expert levels.

### B.1.2. MAMUJOCO DATASET

MaMujoco (de Witt et al., 2020b) is a benchmark for continuous cooperative multi-agent robotic control. Derived from OpenAI Gym's MuJoCo suite, MaMujoco introduces scenarios where multiple agents within a single robot must solve tasks cooperatively. We evaluate on the tasks `Hopper-v2`, `Ant-v2`, and `HalfCheetah-v2`. The offline dataset for MaMujoco was sourced from the work of Xiangsen et al., who used the HAPPO method to train agents. Each task includes datasets with varying quality levels: medium-replay, medium, and expert.

## B.2. LLM-based Preference Annotations

To generate preference annotations for trajectory pairs, we utilized GPT-4o (OpenAI, 2024). This model was prompted with detailed trajectory state information, including key metrics such as health points, shields, relative positions, cooldown times, agent types, and action meanings. The inclusion of such detailed state information significantly improves the ability of the LLM to evaluate trajectory pairs effectively. Following the methodology of DPM (Kang et al., 2024), we extracted critical state details such as the health points of allied and enemy agents, the number of agent deaths (both allied and enemy), and the total remaining health at the final state of each trajectory. These extracted metrics were then used to construct prompts for the LLM, as shown in Table 5.

The OpenAI Batch API (OpenAI, 2025) was employed to submit these prompts to GPT-4o, and the associated token usage and costs are summarized in Table 4. The total cost for generating LLM-based annotations across all tasks was approximately $42, with each dataset containing 2,000 trajectory pairs. While this approach is effective, it becomes costly when scaling to larger datasets or additional tasks.

It is important to note that this method is particularly suited for environments like SMACv1 and SMACv2, where trajectory states provide meaningful and interpretable information. However, the approach has limitations in environments such as MaMujoco, which lack detailed trajectory state information. In MaMujoco tasks, the trajectory states do not include interpretable metrics like health points or agent-specific details, making it infeasible to construct meaningful prompts for LLMs. As a result, only rule-based methods were used to generate preference labels for MaMujoco datasets.

This limitation highlights a broader challenge of the DPM approach (Kang et al., 2024): it relies on the availability of meaningful final state information, which restricts its applicability to specific environments. It is less suitable for long-horizon transitions or environments with image-based observations, where extracting detailed and interpretable state information is either infeasible or computationally expensive.

## B.3. Implementation Details

All experiments were implemented using **PyTorch** and executed in parallel on a single **NVIDIA® H100 NVL Tensor Core GPU** to ensure computational efficiency. We developed two versions of our proposed method, **O-MAPL**, tailored to the specific characteristics of continuous and discrete action domains:

**Continuous Domain (MaMujoco):** For continuous environments, we utilized a *Gaussian distribution* (`torch.distributions.Normal`) to model the policy. Each agent's action is sampled from this distribution, which is parameterized by the mean and standard deviation outputted by the policy network.

**Discrete Domains (SMACv1 & SMACv2):** For discrete environments, we employed a *Categorical distribution* (`torch.distributions.Categorical`) to model the policy. The probability of each action for an agent is computed using the softmax operation over only the *available actions* for that agent. Actions that are not available are assigned a probability of zero. This ensures that the log-likelihood calculation is accurate and avoids penalizing the agent for infeasible actions.

## B.4. Hyperparameters

Table 6 reports hyperparameters used consistently across all experiments:

## B.5. Baseline Comparisons

We compared O-MAPL against four baseline methods to evaluate its performance:

| Tasks | Completion Tokens | Prompt Tokens | Estimated API Cost |
|---|---|---|---|
| 2c_vs_64zg | 5,920 | 2,498,000 | $3.13 |
| 5m_vs_6m | 5,913 | 1,386,000 | $1.74 |
| 6h_vs_8z | 5,941 | 1,462,000 | $1.83 |
| corridor | 5,926 | 1,772,000 | $2.22 |
| protoss_5_vs_5 | 5,920 | 1,460,000 | $1.83 |
| protoss_10_vs_10 | 5,918 | 1,660,000 | $2.08 |
| protoss_10_vs_11 | 5,940 | 1,680,000 | $2.11 |
| protoss_20_vs_20 | 5,901 | 2,060,000 | $2.58 |
| protoss_20_vs_23 | 5,990 | 2,122,000 | $2.66 |
| terran_5_vs_5 | 5,990 | 1,442,000 | $1.81 |
| terran_10_vs_10 | 5,925 | 1,642,000 | $2.06 |
| terran_10_vs_11 | 5,930 | 1,662,000 | $2.08 |
| terran_20_vs_20 | 5,944 | 2,042,000 | $2.56 |
| terran_20_vs_23 | 5,977 | 2,104,000 | $2.64 |
| zerg_5_vs_5 | 5,940 | 1,448,000 | $1.82 |
| zerg_10_vs_10 | 5,914 | 1,648,000 | $2.07 |
| zerg_10_vs_11 | 5,912 | 1,668,000 | $2.09 |
| zerg_20_vs_20 | 5,942 | 2,048,000 | $2.57 |
| zerg_20_vs_23 | 5,913 | 2,110,000 | $2.64 |
| Total | 112,756 | 33,914,000 | $42.53 |

Table 4: GPT-4o API costs

- **BC (Behavior Cloning):** A simple BC supervised learning approach based on preferred trajectories in the dataset.

- **IIPL (Independent Inverse Preference Learning):** Implements IPL (Hejna & Sadigh, 2024) independently for each agent without considering inter-agent coordination.

- **IPL-VDN (Inverse Preference Learning with VDN):** Similar to our O-MAPL algorithm, except that the global Q and V functions are aggregated by summing the local Q-values of individual agents, instead of using a mixing network (Sunehag et al., 2017).

- **SL-MARL (Supervised Learning for MARL):** A two-step approach where the reward function is first learned via supervised learning, followed by policy training through a MARL algorithm (i.e. OMIGA (Wang et al., 2022)), using the learned reward function.

### B.6. Evaluation Metrics

We report two key metrics to assess agent performance:

- **Mean/Standard Deviation of Returns:** Measures the average cumulative rewards achieved by the agents across episodes (applicable to all the environments).

- **Mean/Standard Deviation of Win Rates:** Applicable only to competitive environments (only applicable to SMACv1 and SMACv2). This metric evaluates the percentage of episodes where agents achieve victory.

Each metric is computed as the average and standard deviation of the final results across all four random seeds. Additionally, we present evaluation curves for each method, depicting performance trends during the agent training process using offline datasets.

Prompt

You are a helpful and honest judge of good game playing and progress in the
StarCraft Multi-Agent Challenge game. Always answer as helpfully as possible, while
being truthful.
If you don't know the answer to a question, please don't share false information.
I'm looking to have you evaluate a scenario in the StarCraft Multi-Agent Challenge.
Your role will be to assess how much the actions taken by multiple agents in a given
situation have contributed to achieving victory.

The basic information for the evaluation is as follows.

- Scenario : 5m_vs_6m
- Allied Team Agent Configuration : five Marines(Marines are ranged units in
StarCraft 2).
- Enemy Team Agent Configuration : six Marines(Marines are ranged units in StarCraft
2).
- Situation Description : The situation involves the allied team and the enemy team
engaging in combat, where victory is achieved by defeating all the enemies.
- Objective : Defeat all enemy agents while ensuring as many allied agents as
possible survive.
* Important Notice : You should prefer the trajectory where our allies' health is
preserved while significantly reducing the enemy's health. In similar situations,
you should prefer shorter trajectory lengths.

I will provide you with two trajectories, and you should select the better
trajectory based on the outcomes of these trajectories. Regarding the trajectory, it
will inform you about the final states, and you should select the better case based
on these two trajectories.

[Trajectory 1]
1. Final State Information
    1) Allied Agents Health : 0.000, 0.000, 0.067, 0.067, 0.000
    2) Enemy Agents Health : 0.000, 0.000, 0.000, 0.000, 0.000, 0.040
    3) Number of Allied Deaths : 3
    4) Number of Enemy Deaths : 5
    5) Total Remaining Health of Allies : 0.133
    6) Total Remaining Health of Enemies : 0.040
2. Total Number of Steps : 28

[Trajectory 2]
1. Final State Information
    1) Allied Agents Health : 0.000, 0.000, 0.000, 0.000, 0.000
    2) Enemy Agents Health : 0.120, 0.000, 0.000, 0.000, 0.000, 0.200
    3) Number of Allied Deaths : 5
    4) Number of Enemy Deaths : 4
    5) Total Remaining Health of Allies : 0.000
    6) Total Remaining Health of Enemies : 0.320
2. Total Number of Steps : 23

Your task is to inform which one is better between [Trajectory1] and [Trajectory2]
based on the information mentioned above. For example, if [Trajectory 1] seems
better, output #1, and if [Trajectory 2] seems better, output #2. If it's difficult
to judge or they seem similar, please output #0.
* Important : Generally, it is considered better when fewer allied agents are killed
or injured while inflicting more damage on the enemy.

Omit detailed explanations and just provide the answer.

Table 5: Sample prompt to generate preference data in SMAC environments.

| Hyperparameter | Value |
|---|---|
| Optimizer | Adam |
| Learning rate (Q-value and policy networks) | 1e-4 |
| Tau (soft update target rate) | 0.005 |
| Gamma (discount factor) | 0.99 |
| Batch size | 32 |
| Agent hidden dimension | 256 |
| Mixer hidden dimension | 64 |
| Number of seeds | 4 |
| Number of episodes per evaluation step | 32 |
| Number of evaluation steps | 100 |

Table 6: Hyperparameters used in all experiments.

**Notes on Evaluation:**  For **MaMujoco**, win rates are not applicable as it is not a competitive environment. Evaluation scores are averaged over the results of all four seeds to ensure statistical robustness. The performance trends and comparisons are visualized in detailed figures to provide insights into the training dynamics of each method.

This setup ensures a fair and comprehensive comparison between O-MAPL and the baseline methods in both continuous and discrete multi-agent reinforcement learning tasks.

### B.7. Recovered Rewards

| Tasks | Rule-based | | LLM-based | |
|---|---|---|---|---|
| | Lower | Higher | Lower | Higher |
| 2c_vs_64zg | -8.36±0.26 | 9.25±0.67 | -12.87±0.73 | 14.14±0.80 |
| 5m_vs_6m | -4.49±0.12 | 4.80±0.15 | -4.02±0.20 | 4.51±0.18 |
| 6h_vs_8z | -4.72±0.28 | 5.15±0.22 | -5.11±0.32 | 5.28±0.16 |
| corridor | -12.59±0.31 | 11.23±1.06 | -12.97±0.33 | 10.93±0.45 |
| protoss_5_vs_5 | -6.31±0.22 | 6.54±0.51 | -8.06±0.64 | 7.46±0.77 |
| protoss_10_vs_10 | -7.73±0.18 | 7.92±0.32 | -10.65±1.15 | 9.32±0.91 |
| protoss_10_vs_11 | -7.95±0.69 | 8.31±0.91 | -11.01±0.93 | 10.43±1.57 |
| protoss_20_vs_20 | -8.31±0.35 | 8.19±0.16 | -10.57±0.86 | 9.54±0.74 |
| protoss_20_vs_23 | -8.01±0.22 | 9.10±0.14 | -12.17±0.72 | 12.09±0.80 |
| terran_5_vs_5 | -6.85±0.30 | 6.93±0.56 | -7.85±0.27 | 7.82±0.57 |
| terran_10_vs_10 | -8.25±0.82 | 7.35±0.61 | -10.73±1.49 | 8.16±0.56 |
| terran_10_vs_11 | -8.53±0.67 | 9.62±0.54 | -9.18±0.23 | 10.97±1.38 |
| terran_20_vs_20 | -8.59±0.36 | 8.44±0.22 | -10.44±0.96 | 10.79±1.00 |
| terran_20_vs_23 | -8.49±0.65 | 8.91±0.27 | -14.90±2.06 | 17.95±2.91 |
| zerg_5_vs_5 | -3.74±0.14 | 3.64±0.14 | -5.09±0.19 | 3.51±0.06 |
| zerg_10_vs_10 | -4.16±0.16 | 4.27±0.16 | -5.93±0.43 | 6.14±0.64 |
| zerg_10_vs_11 | -4.54±0.06 | 4.60±0.14 | -7.28±0.50 | 6.20±0.50 |
| zerg_20_vs_20 | -5.31±0.08 | 5.25±0.20 | -7.71±0.54 | 7.24±0.23 |
| zerg_20_vs_23 | -4.78±0.12 | 5.08±0.15 | -8.26±1.13 | 8.00±0.43 |

Table 7: Mean/std recovered rewards of the higher and lower preferred trajectories

Using recovered reward function $R(\mathbf{o}, \mathbf{a}, \mathbf{o}') = \mathcal{M}_\theta[\mathbf{q}(\mathbf{o}, \mathbf{a})] - \gamma\mathcal{M}_\theta[\mathbf{v}(\mathbf{o}')]$, we report the mean/std returns of the higher and lower preferred trajectories in Table 7. Across all tasks, the **higher preferred trajectories** consistently achieve **positive rewards**, while the **lower preferred trajectories** exhibit **negative rewards**. This indicates that the preference-based

learning framework effectively captures and differentiates between preferred and less-preferred trajectories. The consistent separation in rewards suggests that the model successfully aligns policy learning with preference signals, reinforcing high-reward behaviors while penalizing undesirable ones. Moreover, the absolute values of both higher and lower preferred rewards tend to be more extreme in the LLM-based approach, compared to the Rule-based approach. This suggests that LLM-based learning amplifies both positive and negative behaviors, potentially leading to more decisive policy updates, which can be beneficial for clear preference-driven learning.

There are noticeable variations in how different task domains respond to preference-based learning. The **Protoss and Terran** tasks exhibit **larger reward variations**, suggesting that these environments benefit more from preference learning. In contrast, **Zerg tasks** show more moderate reward differences, indicating that either the task dynamics are inherently more balanced or that preference signals have a weaker impact in these settings. Additionally, the **corridor task**, a structured navigation environment, shows similar performance across rule-based and LLM-based approaches.

## B.8. Additional Experimental Details

### B.8.1. RULE-BASED - RETURNS

We present experimental details, in terms of returns, for all tasks (MAMuJoCo, SMACv1, and SMACv2) using rule-based preference datasets.

Table 8 reports the returns and Figure 4 plots the evaluation curves for MaMujoco tasks with Rule-based preference data and Figure The results demonstrate that **O-MAPL** consistently outperforms all baselines across *Hopper-v2*, *Ant-v2*, and *HalfCheetah-v2*, highlighting its effectiveness in rule-based preference learning. Notably, in *Hopper-v2*, O-MAPL achieves a **25.2% higher return** than the next-best method, **SL-MARL**, suggesting superior preference alignment. While SL-MARL performs well in simpler environments, its advantage diminishes in *Ant-v2* and *HalfCheetah-v2*, where **IPL-VDN** shows stronger results. **BC** significantly underperforms, reinforcing the need for preference-based learning over naive imitation.

| Tasks | BC | IIPL | IPL-VDN | SL-MARL | O-MAPL (ours) |
|---|---|---|---|---|---|
| Hopper-v2 | 808.1 ± 39.1 | 782.0 ± 81.5 | 846.6 ± 65.4 | 890.0 ± 88.7 | 1114.4 ± 154.1 |
| Ant-v2 | 1303.9 ± 122.0 | 1312.0 ± 155.6 | 1376.1 ± 142.0 | 1334.1 ± 150.9 | 1406.4 ± 163.7 |
| HalfCheetah-v2 | 4119.9 ± 350.7 | 4028.8 ± 430.0 | 4287.5 ± 273.1 | 4233.9 ± 303.1 | 4382.0 ± 189.7 |

Table 8: Returns for MAMujoco tasks with Rule-based preference data.

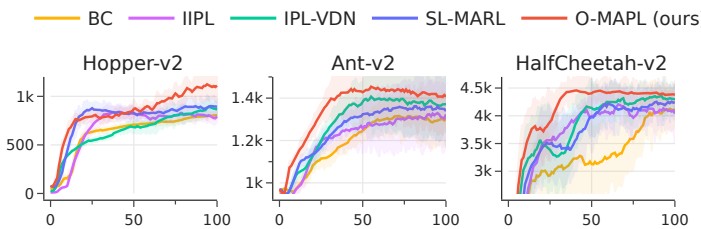

Figure 4: Evaluation curves (returns) for MAMujoco tasks with Rule-based preference data.

Table 9 report the returns and Figure 5 plots the evaluation curves for SMACv1 tasks. The results show that O-MAPL consistently achieves the highest returns across most SMACv1 tasks. While the performance differences are relatively small in simpler tasks like *2c_vs_64zg* and *5m_vs_6m*, O-MAPL outperforms all baselines in more complex scenarios such as *6h_vs_8z*, where it achieves 12.1, compared to the next-best method (SL-MARL, 11.8). Notably, SL-MARL struggles in the *corridor* task, achieving a significantly lower return (14.3) than other methods, suggesting that its reliance on a separate reward modeling phase may be less effective in environments requiring strong coordinated behaviors.

| Tasks | BC | IIPL | IPL-VDN | SL-MARL | O-MAPL (ours) |
|---|---|---|---|---|---|
| 2c_vs_64zg | 19.0 ± 1.1 | 19.3 ± 0.8 | 19.3 ± 1.1 | 19.2 ± 0.8 | 19.3 ± 1.4 |
| 5m_vs_6m | 11.1 ± 2.1 | 10.8 ± 2.0 | 11.2 ± 2.0 | 11.1 ± 2.1 | 11.5 ± 2.1 |
| 6h_vs_8z | 11.0 ± 0.8 | 10.8 ± 0.7 | 11.7 ± 1.0 | 11.8 ± 1.0 | 12.1 ± 1.3 |
| corridor | 19.4 ± 1.0 | 19.4 ± 1.0 | 19.6 ± 1.0 | 14.3 ± 2.8 | 19.6 ± 0.9 |

Table 9: Returns for SMACv1 tasks with Rule-based preference data.

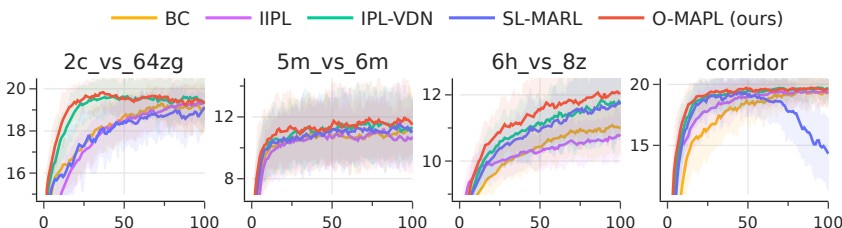

Figure 5: Evaluation curves (returns) for SMACv1 tasks with Rule-based preference data.

Table 10 shows the returns and Figure 6 plots the evaluation curves for SMACv2 tasks (the most complicated ones). The results show that O-MAPL consistently achieves competitive performance across all SMACv2 tasks, often outperforming other baselines. In *Protoss* tasks, O-MAPL achieves the highest returns in most cases, particularly in *protoss_10_vs_10* and *protoss_10_vs_11*, suggesting its effectiveness in complex team-based coordination. Similarly, in *Terran* tasks, O-MAPL consistently outperforms SL-MARL and IPL-VDN, with a noticeable advantage in *terran_20_vs_20* and *terran_20_vs_23*. In *Zerg* environments, O-MAPL continues to show strong results, outperforming all baselines in *zerg_5_vs_5*, *zerg_10_vs_11*, and *zerg_20_vs_20*.

| Tasks | BC | IIPL | IPL-VDN | SL-MARL | O-MAPL (ours) |
|---|---|---|---|---|---|
| protoss_5_vs_5 | 15.4 ± 2.4 | 14.3 ± 2.5 | 17.1 ± 2.7 | 15.8 ± 2.7 | 16.8 ± 2.4 |
| protoss_10_vs_10 | 16.0 ± 2.0 | 15.4 ± 2.1 | 17.8 ± 2.3 | 16.4 ± 2.4 | 17.9 ± 1.9 |
| protoss_10_vs_11 | 12.5 ± 2.3 | 12.7 ± 2.4 | 14.7 ± 2.3 | 14.2 ± 2.2 | 14.9 ± 2.0 |
| protoss_20_vs_20 | 16.7 ± 1.8 | 16.9 ± 1.6 | 18.0 ± 1.5 | 17.3 ± 1.6 | 18.0 ± 1.5 |
| protoss_20_vs_23 | 13.3 ± 2.1 | 13.1 ± 1.8 | 14.9 ± 1.9 | 13.6 ± 1.9 | 14.9 ± 2.0 |
| terran_5_vs_5 | 10.2 ± 2.9 | 11.2 ± 3.1 | 11.9 ± 3.1 | 13.0 ± 3.1 | 12.8 ± 3.5 |
| terran_10_vs_10 | 10.9 ± 2.9 | 10.6 ± 3.0 | 11.6 ± 2.8 | 11.4 ± 2.9 | 11.8 ± 2.6 |
| terran_10_vs_11 | 8.3 ± 2.6 | 8.1 ± 2.3 | 10.1 ± 3.0 | 9.6 ± 2.6 | 11.0 ± 2.8 |
| terran_20_vs_20 | 10.1 ± 2.4 | 10.2 ± 2.5 | 10.7 ± 2.6 | 10.9 ± 2.2 | 11.8 ± 2.4 |
| terran_20_vs_23 | 7.6 ± 2.1 | 7.0 ± 2.1 | 8.7 ± 2.1 | 7.7 ± 2.0 | 9.4 ± 2.0 |
| zerg_5_vs_5 | 11.2 ± 2.8 | 10.5 ± 2.8 | 12.1 ± 2.7 | 12.7 ± 3.1 | 13.1 ± 3.5 |
| zerg_10_vs_10 | 12.9 ± 2.4 | 12.4 ± 2.7 | 13.0 ± 2.6 | 13.2 ± 2.7 | 14.0 ± 2.6 |
| zerg_10_vs_11 | 11.1 ± 2.7 | 10.7 ± 2.7 | 12.1 ± 2.5 | 11.8 ± 2.1 | 12.8 ± 2.7 |
| zerg_20_vs_20 | 13.0 ± 2.2 | 12.2 ± 1.9 | 13.8 ± 2.1 | 12.2 ± 1.6 | 13.9 ± 1.8 |
| zerg_20_vs_23 | 12.1 ± 2.3 | 11.3 ± 1.7 | 12.1 ± 1.8 | 12.2 ± 1.6 | 12.7 ± 2.0 |

Table 10: Returns for SMACv2 tasks with Rule-based preference data.

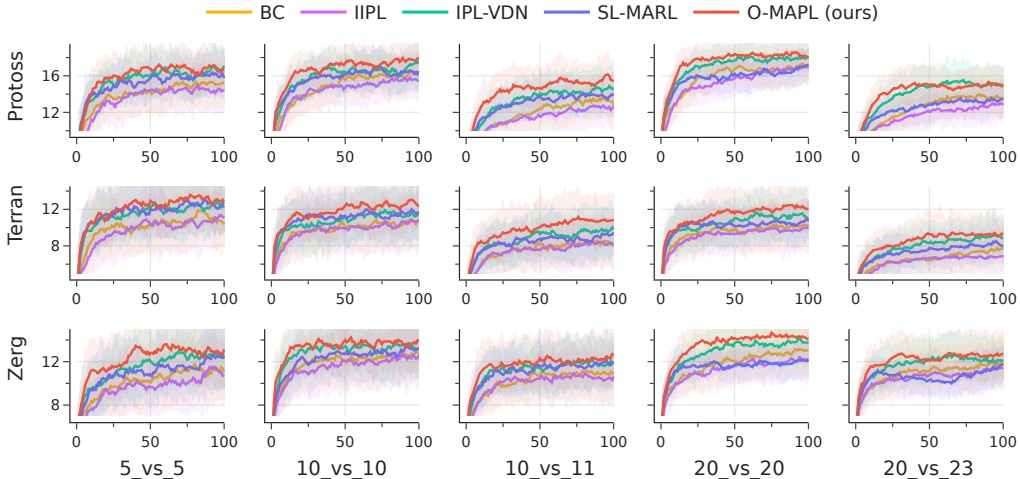

Figure 6: Evaluation curves (returns) for SMACv2 tasks with Rule-based preference data.

### B.8.2. RULE-BASED - WINRATES

For SMACv1 and SMACv2, win rates provide a more meaningful comparison of algorithm performance. The following tables and figures present win rates for SMAC tasks using *rule-based preference data.*

Table 11 shows the winrates and Figure 7 plots the evaluation curves (in terms of winrates) for SMACv1 tasks. The results indicate that O-MAPL consistently achieves the highest winrates across most tasks. In *2c_vs_64zg*, O-MAPL outperforms all baselines, achieving a win rate of 74.4, surpassing IPL-VDN and SL-MARL. Similarly, in *5m_vs_6m* and *6h_vs_8z*, O-MAPL achieves the highest winrates, though the performance gap is less pronounced. Notably, in the *corridor* task, IPL-VDN slightly outperforms O-MAPL, while SL-MARL struggles significantly, indicating that its two-phase approach may be less effective in highly structured navigation tasks. These results suggest that O-MAPL is well-suited for complex coordination tasks, offering robust winrates across diverse SMACv1 environments.

| Tasks | BC | IIPL | IPL-VDN | SL-MARL | O-MAPL (ours) |
|---|---|---|---|---|---|
| 2c_vs_64zg | 59.6 ± 25.0 | 60.4 ± 24.7 | 71.1 ± 22.0 | 63.5 ± 24.0 | 74.4 ± 24.7 |
| 5m_vs_6m | 16.8 ± 18.0 | 14.3 ± 17.0 | 16.8 ± 18.0 | 16.0 ± 18.9 | 19.3 ± 19.6 |
| 6h_vs_8z | 0.6 ± 3.8 | 0.2 ± 2.2 | 2.5 ± 7.6 | 1.6 ± 6.8 | 4.5 ± 11.0 |
| corridor | 89.3 ± 15.5 | 89.8 ± 15.4 | 93.9 ± 11.6 | 49.0 ± 22.8 | 93.2 ± 13.5 |

Table 11: Winrates for SMACv1 tasks with Rule-based preference data.

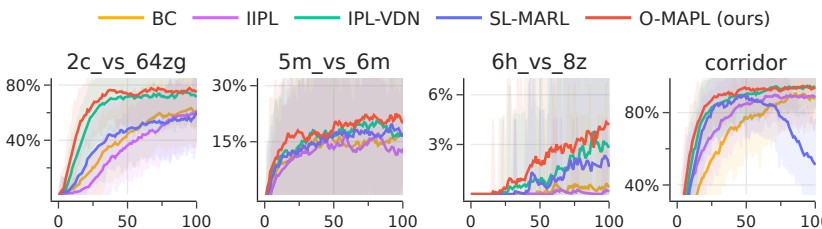

Figure 7: Evaluation curves (in winrates) for SMACv1 tasks with Rule-based preference data.

Table 12 shows the winrates and Figure 8 plots the evaluation curves (in terms of winrates) for SMACv2 tasks. The results, again, show that O-MAPL consistently achieves the highest winrates across most SMACv2 tasks. In the *Protoss* tasks,

O-MAPL outperforms all baselines, particularly in *protoss_10_vs_11* and *protoss_20_vs_20*, where it shows a significant improvement over the other methods. In *Terran* tasks, O-MAPL also achieves the best performance, with notable advantages in *terran_20_vs_20* and *terran_20_vs_23*, where other methods struggle to achieve high winrates. Similarly, in *Zerg* tasks, O-MAPL consistently achieves the best results, particularly in *zerg_20_vs_20* and *zerg_20_vs_23*.

| Tasks | BC | IIPL | IPL-VDN | SL-MARL | O-MAPL (ours) |
|---|---|---|---|---|---|
| protoss_5_vs_5 | 38.1 ± 24.2 | 31.4 ± 25.2 | 54.5 ± 25.9 | 49.0 ± 28.2 | 54.3 ± 24.2 |
| protoss_10_vs_10 | 38.7 ± 24.2 | 28.5 ± 21.8 | 47.9 ± 27.2 | 40.6 ± 23.2 | 53.7 ± 23.6 |
| protoss_10_vs_11 | 12.7 ± 17.4 | 12.5 ± 16.5 | 22.3 ± 21.0 | 18.6 ± 18.8 | 30.7 ± 19.8 |
| protoss_20_vs_20 | 39.8 ± 24.9 | 35.4 ± 21.5 | 57.0 ± 24.8 | 38.7 ± 23.1 | 59.8 ± 23.2 |
| protoss_20_vs_23 | 15.2 ± 18.5 | 9.0 ± 14.2 | 22.7 ± 21.7 | 11.1 ± 14.6 | 23.4 ± 19.2 |
| terran_5_vs_5 | 27.5 ± 24.0 | 26.2 ± 19.5 | 36.3 ± 24.8 | 34.2 ± 23.4 | 39.5 ± 24.7 |
| terran_10_vs_10 | 23.8 ± 20.5 | 21.1 ± 20.8 | 25.8 ± 19.7 | 23.2 ± 19.6 | 28.3 ± 20.6 |
| terran_10_vs_11 | 10.2 ± 15.4 | 7.2 ± 13.3 | 18.2 ± 19.4 | 11.3 ± 15.3 | 18.2 ± 18.7 |
| terran_20_vs_20 | 13.1 ± 17.1 | 11.9 ± 18.2 | 21.5 ± 20.4 | 8.8 ± 13.5 | 23.0 ± 22.4 |
| terran_20_vs_23 | 3.9 ± 10.6 | 4.1 ± 10.3 | 5.7 ± 11.4 | 2.3 ± 7.3 | 7.2 ± 12.9 |
| zerg_5_vs_5 | 23.4 ± 21.1 | 23.6 ± 21.0 | 31.1 ± 20.4 | 33.0 ± 22.5 | 35.2 ± 25.7 |
| zerg_10_vs_10 | 25.8 ± 21.6 | 25.8 ± 22.5 | 32.2 ± 24.6 | 30.7 ± 24.0 | 34.8 ± 22.1 |
| zerg_10_vs_11 | 19.3 ± 20.1 | 12.9 ± 17.4 | 22.5 ± 20.5 | 19.3 ± 18.0 | 23.4 ± 21.1 |
| zerg_20_vs_20 | 19.9 ± 21.0 | 11.1 ± 16.2 | 22.5 ± 21.4 | 5.7 ± 10.9 | 24.8 ± 20.8 |
| zerg_20_vs_23 | 13.1 ± 17.7 | 7.8 ± 12.8 | 12.5 ± 15.3 | 7.6 ± 13.1 | 18.8 ± 18.5 |

Table 12: Winrates for SMACv2 tasks with Rule-based preference data.

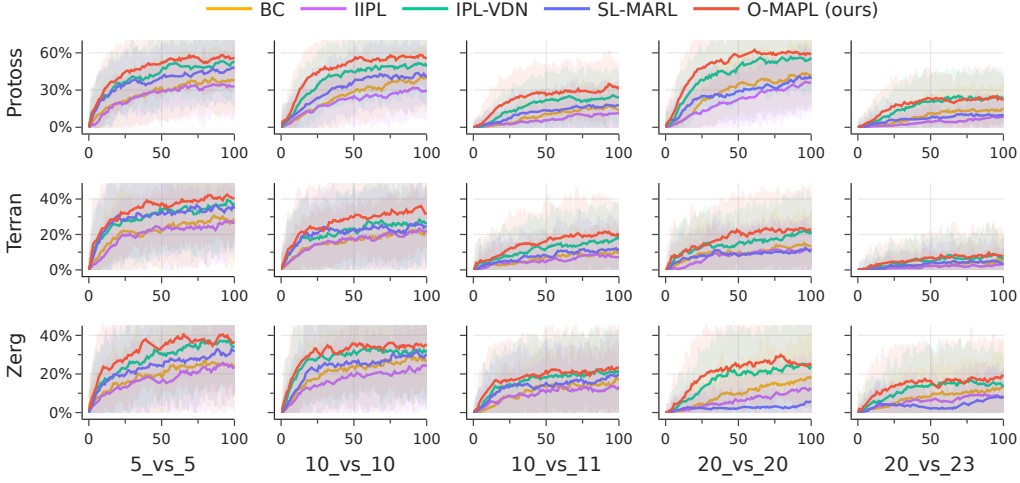

Figure 8: Evaluation curves (in winrates) for SMACv2 tasks with Rule-based preference data.

### B.8.3. LLM-BASED - RETURNS

We present comparisons in terms of returns using LLM-based preference datasets. As noted earlier, only SMACv1 and SMACv2 are suitable for obtaining meaningful preference-based data from LLMs. Therefore, we report comparisons exclusively for SMAC tasks.

Table 13 shows the returns and Figure 9 plots the evaluation curves (in terms of returns) for SMACv1 tasks. The results in Table 13 indicate that O-MAPL consistently achieves the highest or near-highest returns across all SMACv1 tasks using

LLM-based preference data. In *6h_vs_8z*, O-MAPL shows a clear advantage, reaching 12.2, outperforming all baselines. Similarly, in *corridor*, it achieves the highest return (19.7), alongside IPL-VDN, while SL-MARL struggles significantly in this task. Across *2c_vs_64zg* and *5m_vs_6m*, performance differences are minimal, but O-MAPL remains competitive. These results highlight the effectiveness of O-MAPL in leveraging LLM-based preferences, particularly in more complex multi-agent coordination scenarios.

| Tasks | BC | IIPL | IPL-VDN | SL-MARL | O-MAPL (ours) |
|---|---|---|---|---|---|
| 2c_vs_64zg | 19.4 ± 0.9 | 19.3 ± 0.9 | 19.6 ± 1.0 | 19.5 ± 0.7 | 19.6 ± 1.1 |
| 5m_vs_6m | 11.3 ± 2.1 | 10.8 ± 2.0 | 11.4 ± 2.2 | 11.2 ± 2.1 | 11.5 ± 2.3 |
| 6h_vs_8z | 11.1 ± 0.8 | 10.9 ± 0.7 | 11.9 ± 1.1 | 11.8 ± 1.2 | 12.2 ± 1.3 |
| corridor | 19.4 ± 1.0 | 19.4 ± 1.0 | 19.7 ± 0.9 | 15.1 ± 2.4 | 19.7 ± 0.8 |

Table 13: Return comparison for SMACv1 tasks with LLM-based preference data.

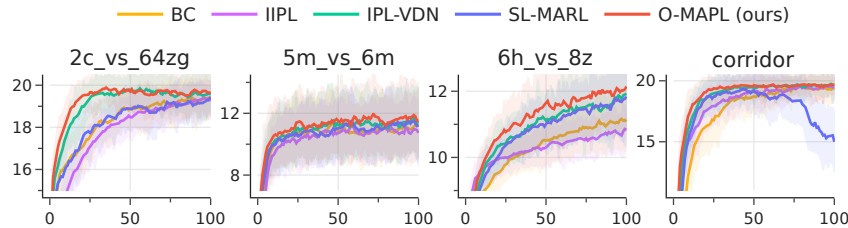

Figure 9: Evaluation curves (in returns) for SMACv1 tasks with LLM-based preference data.

Table 14 shows the returns and Figure 10 plots the evaluation curves (in terms of returns) for SMACv1 tasks. The results demonstrate that O-MAPL consistently achieves the highest returns across most SMACv2 tasks. In *Protoss* tasks, O-MAPL outperforms other methods, particularly in *protoss_10_vs_11* and *protoss_20_vs_20*, indicating its effectiveness in learning structured team-based strategies. In *Terran* tasks, O-MAPL generally achieves the best performance, with notable improvements in *terran_20_vs_20*, suggesting its strength in complex coordination settings. In *Zerg* tasks, O-MAPL maintains strong performance, particularly in *zerg_20_vs_20*, where it achieves the highest return.

| Tasks | BC | IIPL | IPL-VDN | SL-MARL | O-MAPL (ours) |
|---|---|---|---|---|---|
| protoss_5_vs_5 | 16.7 ± 2.7 | 15.9 ± 2.5 | 17.6 ± 2.5 | 16.9 ± 2.4 | 17.9 ± 2.5 |
| protoss_10_vs_10 | 16.5 ± 2.0 | 16.6 ± 2.2 | 17.9 ± 1.8 | 17.5 ± 1.8 | 18.0 ± 2.1 |
| protoss_10_vs_11 | 14.7 ± 2.3 | 14.5 ± 2.0 | 15.4 ± 2.4 | 14.0 ± 2.4 | 16.5 ± 2.2 |
| protoss_20_vs_20 | 17.2 ± 1.7 | 17.6 ± 1.7 | 18.5 ± 1.3 | 18.2 ± 1.9 | 18.9 ± 1.5 |
| protoss_20_vs_23 | 14.3 ± 2.0 | 13.4 ± 1.8 | 15.1 ± 1.8 | 14.3 ± 1.8 | 15.8 ± 1.9 |
| terran_5_vs_5 | 11.8 ± 3.3 | 12.5 ± 3.1 | 13.4 ± 2.9 | 12.6 ± 3.1 | 12.6 ± 2.6 |
| terran_10_vs_10 | 11.3 ± 2.6 | 11.7 ± 3.0 | 11.6 ± 2.7 | 12.1 ± 2.8 | 12.5 ± 2.7 |
| terran_10_vs_11 | 9.2 ± 2.8 | 9.3 ± 2.7 | 10.1 ± 2.6 | 9.9 ± 2.6 | 10.7 ± 2.5 |
| terran_20_vs_20 | 11.2 ± 2.3 | 10.8 ± 2.5 | 11.4 ± 2.4 | 11.6 ± 2.2 | 13.0 ± 2.8 |
| terran_20_vs_23 | 8.5 ± 2.4 | 7.7 ± 2.2 | 8.9 ± 2.1 | 8.7 ± 1.9 | 9.1 ± 2.3 |
| zerg_5_vs_5 | 11.4 ± 2.7 | 11.6 ± 3.0 | 12.8 ± 3.3 | 11.8 ± 2.9 | 12.9 ± 2.6 |
| zerg_10_vs_10 | 13.5 ± 2.6 | 13.4 ± 2.7 | 13.7 ± 2.5 | 13.7 ± 3.0 | 14.5 ± 2.6 |
| zerg_10_vs_11 | 12.0 ± 2.3 | 11.9 ± 2.7 | 11.7 ± 2.0 | 12.8 ± 2.4 | 12.6 ± 2.5 |
| zerg_20_vs_20 | 13.9 ± 2.3 | 13.4 ± 1.9 | 14.6 ± 2.0 | 13.8 ± 2.0 | 15.2 ± 2.4 |
| zerg_20_vs_23 | 12.6 ± 2.0 | 12.1 ± 1.9 | 12.4 ± 2.3 | 12.6 ± 1.9 | 12.4 ± 2.2 |

Table 14: Return comparison for SMACv2 tasks with LLM-based preference data.

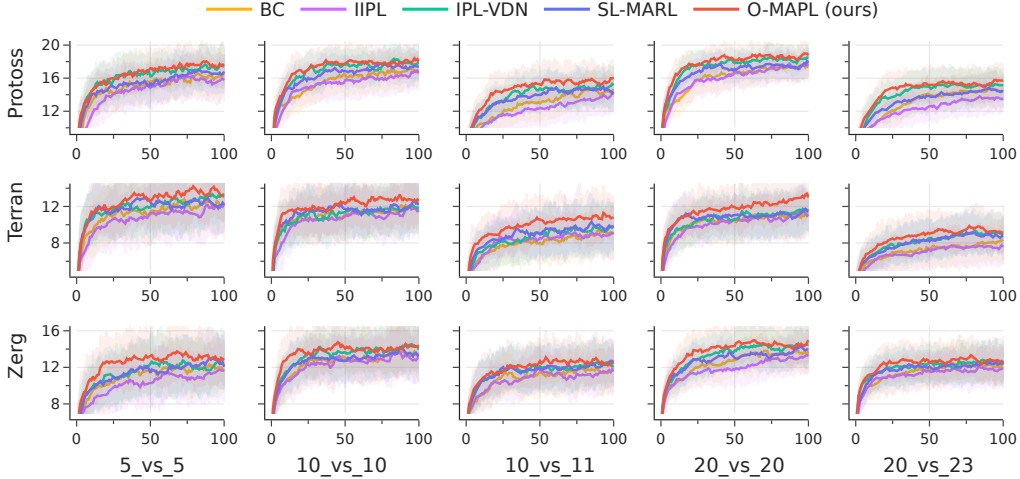

Figure 10: Evaluation curves (in returns) for SMACv2 tasks with LLM-based preference data.

### B.8.4. LLM-BASED - WINRATES

We present a comparison of winrates for SMAC tasks using LLM-based preference datasets.

Table 15 shows the returns and Figure 11 plots the evaluation curves (in terms of winrates) for SMACv1 tasks. The results in Table 15 indicate that O-MAPL achieves the highest win rates across most SMACv1 tasks when using LLM-based preference data. In *2c_vs_64zg*, O-MAPL outperforms all other methods, achieving a win rate of 79.5, slightly higher than IPL-VDN (77.0) and significantly surpassing BC and IIPL. Similarly, in *5m_vs_6m* and *6h_vs_8z*, O-MAPL achieves the best performance, though the overall win rates in these tasks remain low, indicating the increased difficulty of these environments. In the *corridor* task, both O-MAPL and IPL-VDN achieve the highest win rate (94.5), demonstrating their effectiveness in structured navigation tasks, while SL-MARL struggles significantly with a win rate of only 57.6.

| Tasks | BC | IIPL | IPL-VDN | SL-MARL | O-MAPL (ours) |
|---|---|---|---|---|---|
| 2c_vs_64zg | 65.6 ± 24.6 | 60.2 ± 25.9 | 77.0 ± 21.3 | 65.2 ± 21.2 | 79.5 ± 19.6 |
| 5m_vs_6m | 18.2 ± 18.4 | 15.0 ± 17.5 | 18.0 ± 19.2 | 17.4 ± 19.4 | 20.7 ± 20.5 |
| 6h_vs_8z | 0.8 ± 4.3 | 0.4 ± 3.1 | 3.5 ± 9.2 | 3.7 ± 8.9 | 6.1 ± 11.2 |
| corridor | 89.6 ± 15.5 | 90.6 ± 13.6 | 94.5 ± 12.5 | 57.6 ± 22.2 | 94.5 ± 11.2 |

Table 15: Winrate comparison for SMACv1 tasks with LLM-based preference data.

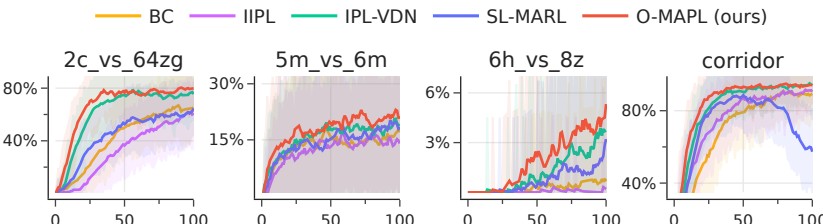

Figure 11: Evaluation curves (in winrates) for SMACv1 tasks with LLM-based preference data.

The results reported in Table 16 and Figure 12 demonstrate that O-MAPL consistently achieves the highest win rates across most SMACv2 tasks using LLM-based preference data. In the *Protoss* tasks, O-MAPL outperforms all baselines, particularly in *protoss_10_vs_11* and *protoss_20_vs_20*, where it achieves substantial improvements over other methods. In *Terran* tasks, O-MAPL also shows strong performance, especially in *terran_20_vs_20* and *terran_20_vs_23*, where other baselines struggle to achieve competitive win rates. In *Zerg* tasks, O-MAPL maintains an advantage, particularly in *zerg_5_vs_5* and *zerg_20_vs_20*, suggesting that its approach generalizes well across different strategic settings. Overall, these results indicate that O-MAPL effectively integrates LLM-based preference data to improve decision-making and coordination in complex multi-agent environments.

| Tasks | BC | IIPL | IPL-VDN | SL-MARL | O-MAPL (ours) |
|---|---|---|---|---|---|
| protoss_5_vs_5 | 48.4 ± 25.9 | 41.0 ± 24.2 | 58.8 ± 24.5 | 54.3 ± 24.0 | 61.5 ± 24.8 |
| protoss_10_vs_10 | 46.3 ± 24.0 | 41.0 ± 24.4 | 57.0 ± 23.4 | 52.5 ± 22.1 | 61.1 ± 24.8 |
| protoss_10_vs_11 | 22.7 ± 22.2 | 15.6 ± 15.9 | 27.3 ± 24.7 | 20.9 ± 20.9 | 34.4 ± 24.8 |
| protoss_20_vs_20 | 48.4 ± 25.3 | 43.6 ± 23.6 | 61.5 ± 22.1 | 51.8 ± 25.0 | 64.5 ± 23.5 |
| protoss_20_vs_23 | 18.0 ± 17.4 | 9.4 ± 14.7 | 23.4 ± 21.4 | 12.1 ± 15.9 | 26.4 ± 20.8 |
| terran_5_vs_5 | 31.1 ± 22.9 | 34.8 ± 23.0 | 41.0 ± 23.7 | 36.7 ± 24.8 | 43.0 ± 23.0 |
| terran_10_vs_10 | 25.8 ± 20.9 | 24.2 ± 21.6 | 32.0 ± 24.4 | 28.9 ± 24.7 | 33.2 ± 23.4 |
| terran_10_vs_11 | 11.7 ± 17.4 | 10.4 ± 15.2 | 17.8 ± 17.7 | 16.4 ± 17.8 | 21.3 ± 20.3 |
| terran_20_vs_20 | 14.5 ± 17.3 | 13.7 ± 17.4 | 21.1 ± 20.4 | 17.2 ± 16.8 | 24.4 ± 23.1 |
| terran_20_vs_23 | 6.4 ± 12.2 | 3.5 ± 9.2 | 7.2 ± 12.6 | 4.7 ± 10.2 | 8.6 ± 14.8 |
| zerg_5_vs_5 | 31.1 ± 22.3 | 26.0 ± 22.2 | 34.8 ± 23.6 | 35.0 ± 23.2 | 40.8 ± 21.6 |
| zerg_10_vs_10 | 31.4 ± 21.9 | 31.1 ± 24.8 | 35.5 ± 23.9 | 33.0 ± 25.0 | 37.9 ± 24.0 |
| zerg_10_vs_11 | 20.1 ± 18.2 | 18.6 ± 20.6 | 22.7 ± 18.3 | 23.0 ± 21.1 | 26.0 ± 23.0 |
| zerg_20_vs_20 | 22.9 ± 21.7 | 16.0 ± 17.3 | 27.3 ± 22.0 | 16.4 ± 18.1 | 31.1 ± 24.6 |
| zerg_20_vs_23 | 15.8 ± 18.5 | 10.4 ± 15.2 | 16.4 ± 19.9 | 13.7 ± 17.4 | 16.0 ± 19.4 |

Table 16: Winrate comparison for SMACv2 tasks with LLM-based preference data.

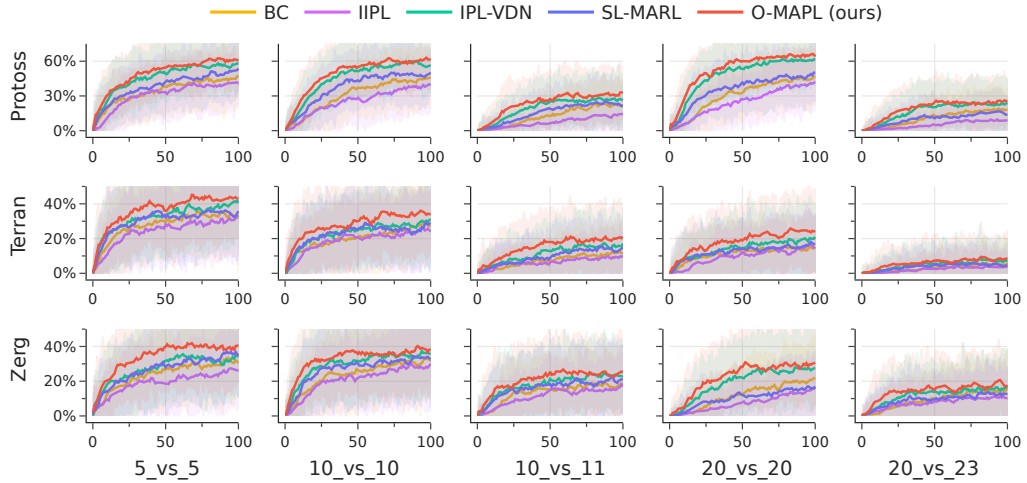

Figure 12: Evaluation curves (in winrates) for SMACv2 tasks with LLM-based preference data.

### B.8.5. IMPACT OF DATA RETENTION

To analyze the effect of reducing the amount of preference data, experiments were conducted by retaining 25%, 50%, 75%, and 100% of the dataset. Both returns and winning rates were evaluated to assess the performance degradation or robustness under reduced data availability. This experiment highlights the sample efficiency of O-MAPL and its ability to generalize with limited data.

| | | **IPL-VDN** | | | | **SL-MARL** | | | | **O-MAPL** (ours) | | | |
|---|---|---|---|---|---|---|---|---|---|---|---|---|---|
| | | 25% | 50% | 75% | 100% | 25% | 50% | 75% | 100% | 25% | 50% | 75% | 100% |
| MaMujoco | Hopper-v2 | 349.4±14.4 | 603.6±74.3 | 706.2±94.0 | 846.6±65.4 | 617.2±53.9 | 701.9±53.3 | 746.0±57.2 | 890.0±88.7 | 628.6±34.8 | 734.5±36.7 | 953.1±75.7 | 1114.4±154.1 |
| | Ant-v2 | 896.3±33.8 | 1141.5±73.1 | 1238.8±135.6 | 1376.1±142.0 | 953.0±32.3 | 1175.2±77.6 | 1281.3±108.0 | 1334.1±150.9 | 1189.3±14.0 | 1282.7±155.6 | 1294.7±167.1 | 1406.4±163.7 |
| | HalfCheetah-v2 | 2676.4±615.4 | 3501.2±336.7 | 3830.0±165.9 | 4287.5±273.1 | 3291.6±402.1 | 3817.0±244.4 | 4022.2±263.1 | 4233.9±303.1 | 3548.3±459.1 | 3943.0±271.3 | 4164.4±277.0 | 4382.0±189.7 |
| SMACv1 | 2c_vs_64zg | 13.4±0.8 | 15.3±1.1 | 16.6±1.0 | 19.3±1.1 | 14.7±1.4 | 16.5±1.1 | 17.4±0.9 | 19.2±0.8 | 16.5±0.9 | 17.3±1.2 | 18.4±1.0 | 19.3±1.4 |
| | 5m_vs_6m | 7.8±2.7 | 9.1±2.2 | 9.9±2.4 | 11.2±2.0 | 8.4±2.3 | 9.6±2.1 | 10.3±1.9 | 11.1±2.1 | 9.5±2.2 | 10.6±2.3 | 11.4±2.0 | 11.5±2.1 |
| | 6h_vs_8z | 7.3±0.7 | 9.0±0.9 | 9.8±0.9 | 11.7±1.0 | 8.3±0.4 | 9.5±0.8 | 10.2±0.9 | 11.8±1.0 | 9.4±0.7 | 10.5±0.9 | 11.0±1.1 | 12.1±1.3 |
| | corridor | 13.2±1.4 | 15.6±0.9 | 16.7±0.8 | 19.6±1.0 | 11.5±1.2 | 12.5±0.6 | 13.6±1.2 | 14.3±2.8 | 16.6±1.0 | 17.6±1.4 | 18.8±0.6 | 19.6±0.9 |
| SMACv2 | protoss 5_vs_5 | 10.9±2.3 | 14.2±2.4 | 14.3±2.3 | 17.1±2.7 | 12.5±2.6 | 12.4±2.0 | 14.6±1.9 | 15.8±2.7 | 13.8±2.2 | 14.1±2.0 | 16.4±2.1 | 16.8±2.4 |
| | 10_vs_10 | 11.5±2.2 | 13.9±1.9 | 15.3±1.3 | 17.8±2.3 | 13.3±2.2 | 15.0±2.0 | 15.1±2.5 | 16.4±2.4 | 14.4±1.7 | 14.8±1.8 | 16.5±1.5 | 17.9±1.9 |
| | 10_vs_11 | 10.6±2.2 | 11.8±1.5 | 13.5±2.0 | 14.7±2.3 | 10.6±1.8 | 10.7±1.7 | 11.6±2.0 | 14.2±2.2 | 12.4±2.2 | 14.5±2.1 | 14.8±2.2 | 14.9±2.0 |
| | 20_vs_20 | 12.6±1.2 | 14.4±1.5 | 16.3±1.6 | 18.0±1.5 | 12.2±1.6 | 15.1±1.8 | 14.5±1.7 | 17.3±1.6 | 15.5±1.5 | 16.7±1.5 | 17.5±1.0 | 18.0±1.5 |
| | 20_vs_23 | 9.6±1.6 | 11.8±1.9 | 12.6±1.5 | 14.9±1.9 | 10.4±2.0 | 11.2±1.8 | 12.5±1.7 | 13.6±1.9 | 12.2±1.6 | 13.4±1.7 | 14.0±1.6 | 14.9±2.0 |
| | terran 5_vs_5 | 8.6±3.1 | 10.1±3.4 | 11.0±3.2 | 11.9±3.1 | 10.6±3.4 | 10.8±2.9 | 10.3±2.8 | 13.0±3.1 | 9.3±2.3 | 11.8±3.6 | 12.5±2.9 | 12.8±3.5 |
| | 10_vs_10 | 8.2±3.5 | 8.7±1.9 | 11.7±3.0 | 11.6±2.8 | 8.0±2.5 | 9.1±2.1 | 10.2±2.7 | 11.4±2.9 | 9.5±2.4 | 10.7±2.9 | 11.4±2.2 | 11.8±2.6 |
| | 10_vs_11 | 7.4±1.9 | 7.6±2.7 | 9.1±2.6 | 10.1±3.0 | 6.7±2.2 | 7.0±1.9 | 7.4±2.2 | 9.6±2.6 | 8.3±2.4 | 9.2±3.0 | 8.7±2.4 | 11.0±2.8 |
| | 20_vs_20 | 7.2±2.1 | 10.3±3.0 | 11.2±2.7 | 10.7±2.6 | 9.3±1.8 | 8.7±2.3 | 10.6±1.8 | 10.9±2.2 | 8.2±2.4 | 9.5±1.7 | 11.2±2.0 | 11.8±2.4 |
| | 20_vs_23 | 5.5±1.7 | 7.0±1.7 | 8.2±2.6 | 8.7±2.1 | 6.4±1.8 | 6.5±2.1 | 7.2±1.8 | 7.7±2.0 | 7.4±1.6 | 8.0±2.2 | 8.3±2.5 | 9.4±2.0 |
| | zerg 5_vs_5 | 7.3±2.5 | 9.4±2.7 | 11.9±3.1 | 12.1±2.7 | 10.0±3.3 | 10.8±2.8 | 12.1±3.2 | 12.7±3.1 | 10.6±3.1 | 12.0±3.2 | 12.4±3.7 | 13.1±3.5 |
| | 10_vs_10 | 9.4±2.9 | 9.4±2.5 | 12.9±2.7 | 13.0±2.6 | 10.6±2.1 | 11.0±2.2 | 12.9±2.3 | 13.2±2.7 | 9.5±2.5 | 11.5±2.2 | 13.5±2.7 | 14.0±2.6 |
| | 10_vs_11 | 8.0±2.4 | 8.9±2.3 | 10.2±2.7 | 12.1±2.5 | 7.9±1.5 | 9.2±2.0 | 11.6±2.2 | 11.8±2.1 | 9.5±2.3 | 10.7±2.3 | 12.4±2.4 | 12.8±2.7 |
| | 20_vs_20 | 9.0±1.9 | 12.9±1.8 | 13.1±2.0 | 13.8±2.1 | 9.0±0.8 | 10.5±1.4 | 11.5±1.8 | 12.2±1.6 | 11.9±1.7 | 13.0±1.6 | 12.0±2.0 | 13.9±1.8 |
| | 20_vs_23 | 9.2±2.1 | 9.8±1.6 | 11.0±1.7 | 12.1±1.8 | 8.9±1.5 | 9.5±1.5 | 9.9±1.7 | 12.2±1.6 | 10.4±1.9 | 10.2±2.2 | 12.5±2.0 | 12.7±2.0 |

Table 17: Returns Comparison with Reduced Preference Rule-based Data

| | | IPL-VDN | | | | SL-MARL | | | | O-MAPL (ours) | | | |
|---|---|---|---|---|---|---|---|---|---|---|---|---|---|
| | | 25% | 50% | 75% | 100% | 25% | 50% | 75% | 100% | 25% | 50% | 75% | 100% |
| SMACv1 | 2c_vs_64zg | 14.4±0.8 | 16.5±0.9 | 17.4±1.0 | 19.6±1.0 | 15.9±1.1 | 17.6±0.5 | 18.5±0.7 | 19.5±0.7 | 17.5±0.8 | 18.0±1.2 | 19.0±1.3 | 19.6±1.1 |
| | 5m_vs_6m | 8.2±2.1 | 9.4±1.8 | 10.0±2.0 | 11.4±2.2 | 8.8±1.9 | 10.2±2.4 | 11.0±2.2 | 11.2±2.1 | 9.4±1.6 | 11.4±2.0 | 11.8±2.6 | 11.5±2.3 |
| | 6h_vs_8z | 7.8±0.7 | 9.4±1.0 | 10.3±0.9 | 11.9±1.1 | 8.7±0.6 | 10.1±0.9 | 10.8±0.7 | 11.8±1.2 | 10.0±0.9 | 11.1±1.3 | 12.1±1.4 | 12.2±1.3 |
| | corridor | 14.3±1.1 | 16.4±1.2 | 17.5±1.0 | 19.7±0.9 | 12.8±1.0 | 13.6±1.2 | 14.6±1.3 | 15.1±2.4 | 17.1±0.9 | 18.6±0.6 | 19.3±1.1 | 19.7±0.8 |
| SMACv2 | protoss 5_vs_5 | 11.6±2.2 | 14.8±2.7 | 15.9±2.3 | 17.6±2.5 | 13.8±2.1 | 12.8±1.7 | 17.0±2.1 | 16.9±2.4 | 14.9±2.3 | 16.3±1.9 | 17.8±2.0 | 17.9±2.5 |
| | 10_vs_10 | 12.5±1.7 | 15.2±2.0 | 15.1±1.6 | 17.9±1.8 | 13.4±1.8 | 14.7±2.1 | 16.0±2.0 | 17.5±1.8 | 14.8±1.8 | 16.9±1.9 | 16.9±1.5 | 18.0±2.1 |
| | 10_vs_11 | 10.5±1.6 | 13.8±2.5 | 14.5±2.1 | 15.4±2.4 | 11.9±1.8 | 14.0±2.3 | 13.8±1.9 | 14.0±2.4 | 13.6±2.5 | 14.5±1.8 | 15.9±2.0 | 16.5±2.2 |
| | 20_vs_20 | 12.8±1.4 | 14.2±1.3 | 17.8±1.4 | 18.5±1.3 | 14.1±1.2 | 16.2±1.4 | 15.8±1.8 | 18.2±1.9 | 16.7±1.7 | 17.5±1.5 | 18.2±1.3 | 18.9±1.5 |
| | 20_vs_23 | 10.9±2.1 | 12.9±2.6 | 14.5±1.8 | 15.1±1.8 | 10.8±1.6 | 12.7±1.8 | 14.1±1.8 | 14.3±1.8 | 13.0±2.1 | 14.5±2.2 | 15.5±1.9 | 15.8±1.9 |
| | terran 5_vs_5 | 11.8±3.3 | 12.6±3.9 | 12.4±3.0 | 13.4±2.9 | 10.5±2.8 | 12.1±3.2 | 13.3±3.0 | 12.6±3.1 | 10.5±2.4 | 12.0±2.9 | 12.3±2.9 | 12.6±2.6 |
| | 10_vs_10 | 8.4±2.6 | 10.2±2.8 | 10.7±3.2 | 11.6±2.7 | 10.2±2.6 | 9.9±2.3 | 11.0±2.5 | 12.1±2.8 | 9.7±3.0 | 11.2±2.6 | 11.4±2.6 | 12.5±2.7 |
| | 10_vs_11 | 7.5±3.2 | 8.5±2.9 | 8.4±2.3 | 10.1±2.6 | 8.0±2.3 | 8.0±2.6 | 8.5±1.7 | 9.9±2.6 | 8.2±2.0 | 9.6±2.9 | 10.5±2.4 | 10.7±2.5 |
| | 20_vs_20 | 8.7±2.0 | 10.2±2.2 | 11.0±2.5 | 11.4±2.4 | 8.6±1.7 | 9.8±2.1 | 9.6±1.8 | 11.6±2.2 | 9.1±1.6 | 10.4±2.0 | 12.8±2.6 | 13.0±2.8 |
| | 20_vs_23 | 5.8±2.2 | 7.3±2.4 | 8.2±2.1 | 8.9±2.1 | 6.8±1.7 | 7.6±2.2 | 7.6±2.4 | 8.7±1.9 | 8.1±2.1 | 8.7±1.5 | 9.0±2.1 | 9.1±2.3 |
| | zerg 5_vs_5 | 7.6±2.4 | 10.4±2.3 | 10.6±2.3 | 12.8±3.3 | 10.2±2.9 | 11.6±2.5 | 11.0±2.1 | 11.8±2.9 | 11.7±2.8 | 11.5±3.4 | 11.8±2.0 | 12.9±2.6 |
| | 10_vs_10 | 9.5±2.7 | 12.1±2.3 | 12.0±2.0 | 13.7±2.5 | 11.1±1.9 | 12.0±2.6 | 12.3±2.4 | 13.7±3.0 | 12.9±1.8 | 13.4±2.9 | 13.2±2.3 | 14.5±2.6 |
| | 10_vs_11 | 8.5±2.0 | 10.2±2.2 | 10.3±1.9 | 11.7±2.0 | 10.4±2.1 | 10.2±2.5 | 12.0±2.3 | 12.8±2.4 | 11.2±2.5 | 11.0±1.8 | 11.0±2.0 | 12.6±2.5 |
| | 20_vs_20 | 9.6±2.3 | 13.3±2.2 | 13.5±1.9 | 14.6±2.0 | 10.4±1.8 | 11.1±1.4 | 11.8±1.5 | 13.8±2.0 | 12.4±1.8 | 12.8±1.4 | 15.0±1.8 | 15.2±2.4 |
| | 20_vs_23 | 9.5±1.8 | 10.4±2.2 | 11.4±1.9 | 12.4±2.3 | 9.5±1.7 | 11.2±1.8 | 11.4±1.6 | 12.6±1.9 | 10.3±1.6 | 11.7±1.7 | 11.7±2.0 | 12.4±2.2 |

Table 18: Returns Comparison with Reduced Preference LLM-based Data

### B.8.6. COMPARISON OF 1-LAYER VS. 2-LAYER MIXERS

This experiment evaluates the impact of using 1-layer linear mixers versus 2-layer non-linear mixers in the value decomposition process. The comparison is carried out on both returns and winning rates for SMACv1 and SMACv2 tasks, using both rule-based and LLM-based datasets. The goal is to validate the theoretical claim that 1-layer mixers provide better convexity and stability, while examining whether 2-layer mixers offer any practical advantage.

| | | Returns | | | | Winning Rates | | | |
|---|---|---|---|---|---|---|---|---|---|
| | | Rule-based | | LLM-based | | Rule-based | | LLM-based | |
| | | 1-layer | 2-layer | 1-layer | 2-layer | 1-layer | 2-layer | 1-layer | 2-layer |
| SMACv1 | 2c_vs_64zg | 19.3±1.4 | 12.9±1.4 | 19.6±1.1 | 13.7±1.6 | 74.4±24.7 | 2.3±7.3 | 79.5±19.6 | 3.9±11.0 |
| | 5m_vs_6m | 11.5±2.1 | 7.1±0.8 | 11.5±2.3 | 7.5±0.5 | 19.3±19.6 | 0.8±4.3 | 20.7±20.5 | 0.0±0.0 |
| | 6h_vs_8z | 12.1±1.3 | 9.5±0.4 | 12.2±1.3 | 9.9±0.6 | 4.5±11.0 | 0.0±0.0 | 6.1±11.2 | 0.0±0.0 |
| | corridor | 19.6±0.9 | 4.4±1.4 | 19.7±0.8 | 2.7±0.7 | 93.2±13.5 | 0.0±0.0 | 94.5±11.2 | 0.0±0.0 |
| SMACv2 | protoss 5_vs_5 | 16.8±2.4 | 13.8±2.7 | 17.9±2.5 | 16.3±2.9 | 54.3±24.2 | 36.7±25.8 | 61.5±24.8 | 48.4±29.3 |
| | 10_vs_10 | 17.9±1.9 | 15.5±1.7 | 18.0±2.1 | 16.7±1.9 | 53.7±23.6 | 42.9±20.1 | 61.1±24.8 | 29.7±19.2 |
| | 10_vs_11 | 14.9±2.0 | 13.7±2.1 | 16.5±2.2 | 13.3±2.1 | 30.7±19.8 | 14.1±16.2 | 34.4±24.8 | 21.1±17.8 |
| | 20_vs_20 | 18.0±1.5 | 15.3±1.7 | 18.9±1.5 | 15.6±1.6 | 59.8±23.2 | 23.4±18.7 | 64.5±23.5 | 29.7±24.6 |
| | 20_vs_23 | 14.9±2.0 | 12.5±1.6 | 15.8±1.9 | 13.5±1.6 | 23.4±19.2 | 9.8±16.5 | 26.4±20.8 | 11.7±16.5 |
| | terran 5_vs_5 | 12.8±3.5 | 10.2±2.9 | 12.6±2.6 | 11.2±2.7 | 39.5±24.7 | 26.0±20.8 | 43.0±23.0 | 22.7±20.1 |
| | 10_vs_10 | 11.8±2.6 | 9.0±2.6 | 12.5±2.7 | 10.0±2.4 | 28.3±20.6 | 19.4±23.0 | 33.2±23.4 | 26.7±22.0 |
| | 10_vs_11 | 11.0±2.8 | 8.1±2.4 | 10.7±2.5 | 9.0±3.1 | 18.2±18.7 | 7.0±14.3 | 21.3±20.3 | 13.3±20.7 |
| | 20_vs_20 | 11.8±2.4 | 9.4±2.0 | 13.0±2.8 | 11.9±2.4 | 23.0±22.4 | 12.0±16.5 | 24.4±23.1 | 10.9±15.2 |
| | 20_vs_23 | 9.4±2.0 | 8.0±2.0 | 9.1±2.3 | 7.3±1.4 | 7.2±12.9 | 5.5±10.3 | 8.6±14.8 | 1.6±6.1 |
| | zerg 5_vs_5 | 13.1±3.5 | 10.1±3.7 | 12.9±2.6 | 9.4±3.0 | 35.2±25.7 | 30.6±30.6 | 40.8±21.6 | 25.6±24.4 |
| | 10_vs_10 | 14.0±2.6 | 10.5±3.0 | 14.5±2.6 | 11.2±2.8 | 34.8±22.1 | 24.6±28.5 | 37.9±24.0 | 27.5±27.1 |
| | 10_vs_11 | 12.8±2.7 | 11.9±1.9 | 12.6±2.5 | 11.7±1.8 | 23.4±21.1 | 15.6±16.2 | 26.0±23.0 | 10.9±13.9 |
| | 20_vs_20 | 13.9±1.8 | 11.0±1.4 | 15.2±2.4 | 12.2±1.7 | 24.8±20.8 | 12.0±15.3 | 31.1±24.6 | 12.5±14.0 |
| | 20_vs_23 | 12.7±2.0 | 10.8±1.4 | 12.4±2.2 | 9.9±1.5 | 18.8±18.5 | 3.1±8.3 | 16.0±19.4 | 3.9±9.1 |

Table 19: Comparison of our O-MAPL with 1-layer vs. 2-layer Mixers: Returns and Winning Rates (in percentage)

### B.8.7. COMPARISON OF GPT-4O VS. GPT-4O-MINI

To evaluate the effect of preference data quality, the performance of O-MAPL is compared when using high-quality LLM-generated preferences (GPT-4o) versus lower-quality preferences (GPT-4o-mini). Both returns and winning rates are analyzed to understand the sensitivity of O-MAPL to the quality of preference annotations.

| | | Returns | | Winning Rates | |
|---|---|---|---|---|---|
| | | **4o** | **4o-mini** | **4o** | **4o-mini** |
| SMACv1 | 2c_vs_64zg | 19.6±1.1 | 17.0±1.4 | 79.5±19.6 | 62.8±17.4 |
| | 5m_vs_6m | 11.5±2.3 | 9.6±2.2 | 20.7±20.5 | 19.4±20.2 |
| | 6h_vs_8z | 12.2±1.3 | 10.8±1.2 | 6.1±11.2 | 2.8±8.3 |
| | corridor | 19.7±0.8 | 17.9±0.5 | 94.5±11.2 | 88.1±8.1 |
| SMACv2 | protoss | | | | |
| | 5_vs_5 | 17.9±2.5 | 15.6±2.4 | 61.5±24.8 | 58.1±28.8 |
| | 10_vs_10 | 18.0±2.1 | 17.5±2.4 | 61.1±24.8 | 54.4±28.2 |
| | 10_vs_11 | 16.5±2.2 | 14.2±2.5 | 34.4±24.8 | 29.1±22.8 |
| | 20_vs_20 | 18.9±1.5 | 17.0±1.8 | 64.5±23.5 | 60.0±27.6 |
| | 20_vs_23 | 15.8±1.9 | 15.2±2.6 | 26.4±20.8 | 26.2±30.9 |
| | terran | | | | |
| | 5_vs_5 | 12.6±2.6 | 14.3±3.8 | 43.0±23.0 | 45.9±28.8 |
| | 10_vs_10 | 12.5±2.7 | 9.6±3.3 | 33.2±23.4 | 30.0±30.9 |
| | 10_vs_11 | 10.7±2.5 | 9.8±2.9 | 21.3±20.3 | 13.1±18.5 |
| | 20_vs_20 | 13.0±2.8 | 11.7±2.5 | 24.4±23.1 | 20.6±26.9 |
| | 20_vs_23 | 9.1±2.3 | 7.7±1.9 | 8.6±14.8 | 1.9±11.6 |
| | zerg | | | | |
| | 5_vs_5 | 12.9±2.6 | 11.2±3.4 | 40.8±21.6 | 38.3±28.8 |
| | 10_vs_10 | 14.5±2.6 | 12.7±2.4 | 37.9±24.0 | 34.7±25.2 |
| | 10_vs_11 | 12.6±2.5 | 11.0±2.8 | 26.0±23.0 | 22.5±23.6 |
| | 20_vs_20 | 15.2±2.4 | 12.9±2.2 | 31.1±24.6 | 25.3±25.2 |
| | 20_vs_23 | 12.4±2.2 | 10.2±2.2 | 16.0±19.4 | 14.2±16.6 |

Table 20: Comparison of our O-MAPL with GPT-4o vs. GPT-4o-mini: Returns and Winning Rates (in percentage)

