# OpenReview forum: "O-MAPL: Offline Multi-agent Preference Learning"
_ICML.cc/2025/Conference — ICML 2025 poster_

### Official Review · Reviewer_XeUY · 2025-03-07

**Overall Recommendation:** 4

**Summary:**

This paper studies the problem of cooperative multi-agent reinforcement learning from preference data. The authors formulate the problem as cooperative Markov game with a global reward function where the goal of the agents is to learn optimal policies given offline pairs of trajectories with corresponding preferences. The authors make use of a one-to-one correspondence between the inverse Bellman operator of KL-regularized Q-functions and the global reward function to formulate the optimal policy objective--thereby reducing the problem to learning the optimal Q-functions. The combinatorial computational complexity due to the presence of many agents is tackled via value decomposition and mixing through linear (learnable) maps. Moreover, the problem of maintaining a local-to-global consistency while simultaneously having well-defined policies over probability simplexes is tackled through local weighted behavior cloning techniques. The authors also theoretically demonstrate the correspondence of global optimal policies with the learned local policies and the validity of the returned value functions. Extensive experimental results showcase the benefit of using their proposed method, O-MAPL.

**Claims And Evidence:**

The convexity of the parameter space over the value functions is theoretically proven. Moreover, the consistency between optimal global policies and optimal local policies, in the WBC sense, is also theoretically shown. The validity of O-MAPL algorithm is demonstrated by extensive experiments on several MARL benchmarks against several baselines.

**Essential References Not Discussed:**

To the best of my knowledge, there aren't any essential references missing. The related work section seems adequately comprehensive.

**Experimental Designs Or Analyses:**

I have not checked the validity of the experimental results.

**Methods And Evaluation Criteria:**

The proposed methods and the evaluation criteria are reasonable for the problem at hand.

**Other Comments Or Suggestions:**

N/A

**Other Strengths And Weaknesses:**

1. I really enjoyed reading the paper as it is well-written, clear and coherent.
2. The proposed method has several novel components that are brought together: (i) the reduction of the reward learning problem to the Q-space; (ii) the usage of linear mixing networks for the value decomposition phase; (iii) the usage of local weighted behavior cloning to align local policies to learned global value functions.
3. The utilized rationale behind O-MAPL is justified from theoretical results.
4. The experimental validation of O-MAPL is extensive, which make the algorithm quite practical.

I would have liked to see some simple convergence result which sheds some light into how close the output global policy is to the ground-truth global optimal policy with respect to the global reward function, in terms of problem-specific parameters such as data size, data coverage, policy class and the used mixing networks and WBC technique.

**Questions For Authors:**

1. Why do you switch from discounted MDPs to finite-horizon MDPs when you discuss preference-based learning in Section 4.1? Isn't it better to have a unified framework?
2. Related to my comment on the convergence, is it possible to add some convergence guarantees of O-MAPL? There is an alternating optimization problem being solved for both the optimal Q-function and value function. Due to convexity/concavity, I expect it to converge in finite time. However, what is not clear to me is how the presence of WBC affects convergence.
3. How does data coverage affect the performance of O-MAPL, that is, have you tested it in datasets where not enough "good" trajectories are present? How does it affect the performance?
4. Can you comment on the presence of the additional term in Proposition 4.5? The result indicates that the value function is not "exactly" expressed as log-sum-exp of the Q-function, but there is an additional bias. Why is that?

**Relation To Broader Scientific Literature:**

There are only two papers that discuss MARL from preference data, namely, (Kang et al., 2024) and (Zhang et al., 2024), both of which go through the reward learning phase and then perform policy optimization on top. The approach of the current paper is novel, in that regularized logistic regression is used to obtain estimates of the inverse Bellman operator of the Q-function, which correspond to the logits of the optimal policy directly. Such an approach bypasses the policy optimization using estimated reward functions, which is where it diverges from previous work.

**Theoretical Claims:**

I skimmed through the proofs of the main results in the appendix, and, they seem overall fine. No blunders or unexpected claims. However, I haven't checked in detail.

---

> ### Author Rebuttal · Authors · 2025-03-31
>
> We sincerely appreciate the reviewer for carefully reading our paper, offering a positive evaluation, and providing valuable and insightful comments. To address your questions, we have conducted additional experiments, which are detailed in the **PDF** available via this anonymous link: https://1drv.ms/b/s!AgChHLa7t5Bza5QJpMfi7YJX6PI
>
> ---
>
> Below, we address each of your comments and questions:
>
> > Why do you switch from discounted MDPs to finite-horizon MDPs when you discuss preference-based learning in Section 4.1? Isn't it better to have a unified framework?
>
> We sincerely thank the reviewer for this insightful comment. We initially adopted the discounted infinite-horizon MDP framework as it is widely utilized in RL and MARL literature due to its general applicability. However, for our objective of the preference-based learning, we chose to exclude the discount factor because the preference data commonly consists of partial or sub-trajectory information rather than complete trajectories. Therefore, our algorithm was developed around an undiscounted sum of rewards, which aligns closely with recent preference-based approaches in the single-agent setting. We will clearly articulate and emphasize this choice in the revised manuscript.
>
> > Related to my comment on the convergence, is it possible to add some convergence guarantees of O-MAPL? There is an alternating optimization problem being solved for both the optimal Q-function and value function. Due to convexity/concavity, I expect it to converge in finite time.
>
> We sincerely thank the reviewer for this insightful comment. Our alternating optimization approach is based on reformulating the primary learning problem as an equivalent bilevel optimization problem, as shown below:
>
>  $max_q L(q,v,w)$ s.t. $v= \arg\min  J(v)$
>
> In this formulation, $L(q,v,w)$ is concave in $q$ and  $J(v)$ and is convex in $v$. This convex-concave structure allows the alternating optimization process — where $q$ and $v$ are updated aternatively — to converge to a stationary point. This optimization method is widely adopted and well-supported in the literature on *bilevel optimization*.
>
> We will clarify this point and cite relevant works on bilevel optimization to support our approach. Thank you for bringing this to our attention.
>
> > However, what is not clear to me is how the presence of WBC affects convergence.
>
> We thank the reviewer for this insightful question. To clarify, the convergence of the Q and V functions arises from solving the bilevel optimization problem using an alternating procedure. The convergence of the WBC component, on the other hand, is theoretically supported by Theorem 4.3 under the assumption that Q and V are fixed. In practice, we train the WBC component simultaneously with Q and V updates to improve computational efficiency. Empirically, one can show that as Q and V converge, the WBC also reliably converges to the optimal policy given a sufficient number of update steps.
>
> We will better clarify this point in the revised manuscript.
>
> > How does data coverage affect the performance of O-MAPL, that is, have you tested it in datasets where not enough "good" trajectories are present? How does it affect the performance?
>
> We thank the reviewer for this insightful question. To address this point, we conducted additional experiments to evaluate the effect of data quality and coverage on O-MAPL’s performance. Specifically, we varied the number of preference samples, and examined cases where LLMs provided lower-quality preference labels. The additional results are reported in **Tables 1, 2 and 4** in the **PDF**. These additional results show that O-MAPL remains robust across a range of dataset qualities, although its performance degrades gracefully as the quality of preference data decreases.
>
>  We will include these additional results in the revised manuscript to highlight this robustness.
>
> > Can you comment on the presence of the additional term in Proposition 4.5? The result indicates that the value function is not "exactly" expressed as log-sum-exp of the Q-function, but there is an additional bias. Why is that?
>
> We thank the reviewer for this insightful question. In Proposition 4.5, we demonstrate that the local value function *cannot be exactly expressed* as the log-sum-exp of the local function. This highlights a key limitation of prior MARL approaches that rely solely on local functions to compute local policies or values. The fundamental reason is the interdependence among agents in cooperative settings — local policies are influenced by the joint behavior of other agents. As such, expressing them purely in terms of local and functions leads to an approximation error. We will clarify this point more explicitly in the revised manuscript.
>
> ---
>
> *We hope the above responses satisfactorily address your questions and concerns. If you have any further questions or comments, we would be happy to provide additional clarification.*

---

### Official Review · Reviewer_5DBX · 2025-03-13

**Overall Recommendation:** 3

**Summary:**

This paper introduces O-MAPL, an end-to-end preference-based reinforcement learning framework for cooperative multi-agent systems, addressing the challenge of inferring reward functions from demonstrations in complex MARL settings. Prior methods often separate reward learning and policy optimization, leading to instability. The authors propose a novel approach that bypasses explicit reward modeling by leveraging the relationship between rewards and soft Q-functions in MaxEnt RL.

**Claims And Evidence:**

While the paper argues that prior methods suffer from instability due to phased training, no empirical evidence is provided to demonstrate that O-MAPL actually achieves greater stability. For instance, there are no comparisons of training curves, convergence rates, or sensitivity analyses against baselines like IIPL or IPL-VDN that decouple reward and policy learning. Without such comparisons, the claim remains speculative.

The experiments use LLM-generated preference data for training O-MAPL but do not clarify whether baseline methods (e.g., IIPL, BC) were evaluated under the same data conditions. Performance gains could stem from the quality or scale of LLM-generated data rather than the algorithm itself. Additionally, there is no ablation study comparing rule-based vs. LLM-generated preferences for O-MAPL, making it impossible to disentangle the contribution of the framework from the data source.

**Essential References Not Discussed:**

In my point of view, the related works that are essential to understanding the (context for) key contributions of the paper were all currently cited/discussed in the paper.

**Experimental Designs Or Analyses:**

The experiments use LLM-generated preference data for training O-MAPL but do not clarify whether baseline methods (e.g., IIPL, BC) were evaluated under the same data conditions. Performance gains could stem from the quality or scale of LLM-generated data rather than the algorithm itself. Additionally, there is no ablation study comparing rule-based vs. LLM-generated preferences for O-MAPL, making it impossible to disentangle the contribution of the framework from the data source.

**Methods And Evaluation Criteria:**

The methods are well aligned with the challenges of multi-agent preference-based RL. By integrating MaxEnt RL with value decomposition under CTDE, O-MAPL addresses the instability of prior two-phase approaches while ensuring global-local policy consistency. The use of linear mixing networks and weighted BC for policy extraction is pragmatic for offline settings, balancing stability and scalability.

Evaluation on SMAC (discrete) and MAMuJoCo (continuous) benchmarks covers diverse MARL scenarios, and the inclusion of LLM-generated preference data reflects real-world applicability. While human-labeled datasets are absent (a common MARL limitation), the rule-based/LLM-generated data strategy is reasonable and reproducible. Baselines are relevant, though broader comparisons (e.g., with recent offline MARL methods) could strengthen claims. Overall, the design and evaluation credibly validate the framework’s efficacy.

**Other Comments Or Suggestions:**

None.

**Other Strengths And Weaknesses:**

**Strengths**

1. The paper introduces a novel end-to-end preference-based learning framework for multi-agent RL that bypasses explicit reward modeling by leveraging the intrinsic relationship between reward functions and soft Q-functions. This approach is original, as it creatively combines MaxEnt RL principles with a multi-agent value decomposition strategy, addressing the instability of prior two-phase methods while ensuring global-local policy consistency through theoretical guarantees.

2. The authors rigorously evaluate their method across diverse benchmarks (SMAC, MAMuJoCo) using both rule-based and LLM-generated preference data. The results demonstrate consistent superiority over baselines, particularly highlighting the effectiveness of LLM-based preference data, which opens new avenues for cost-effective training in complex multi-agent systems.

**Weaknesses**

1. The work focuses exclusively on cooperative settings, leaving mixed cooperative-competitive environments unexplored. This restricts the framework’s applicability to real-world scenarios where adversarial interactions or heterogeneous objectives are common, and the proposed value decomposition may not generalize.

2. While leveraging LLMs mitigates data generation costs, the method still requires substantial trajectory pairs for training. The paper does not address sample efficiency in low-data regimes or human-in-the-loop settings, limiting its practicality where preference queries are expensive or scarce.

**Questions For Authors:**

1. The paper emphasizes the use of linear mixing networks to preserve convexity (Prop. 4.1–4.2) but acknowledges that non-linear mixing (e.g., two-layer networks) is common in prior MARL works. However, the experiments do not include comparisons with non-linear mixing variants. Could the authors provide ablation studies or empirical evidence showing that linear mixing indeed outperforms non-linear alternatives in their framework?


2. The LLM-based preference generation is a novel contribution, but the paper does not analyze the quality of LLM-generated labels (e.g., alignment with ground-truth human preferences or rule-based labels). How sensitive is O-MAPL to potential inaccuracies or biases in LLM-generated preferences?

**Relation To Broader Scientific Literature:**

The paper’s key contributions are positioned within the broader MARL literature by advancing offline preference learning and stability in decentralized value decomposition, addressing gaps in prior work while building on established frameworks.

**Theoretical Claims:**

All the proofs are mathematically sound under idealized assumptions

---

> ### Author Rebuttal · Authors · 2025-03-31
>
> We greatly appreciate the reviewer's detailed feedback and constructive questions. To address your questions, we have conducted additional experiments, which are detailed in the **PDF** available via this anonymous link: https://1drv.ms/b/s!AgChHLa7t5Bza5QJpMfi7YJX6PI
>
> ----
>
> Below, we respond point-by-point to your valuable suggestions and concerns.
>
> > There are no comparisons of training curves, convergence rates, or sensitivity analyses against baselines like IIPL or IPL-VDN …
>
> We thank the reviewer for highlighting this important point. To clarify, the curves presented in our paper (Figure 1 and additional figures in the appendix) represent **evaluations during training progress**. Specifically, these curves illustrate the evolution of the value function during training, comparing different approaches (O-MAPL, IIPL, IPL-VDN).
>
> > The experiments use LLM-generated preference data for training O-MAPL but do not clarify whether baseline methods (e.g., IIPL, BC) were evaluated under the same data conditions.
>
> We thank the reviewer for the insightful comment. We would like to clarify that all baselines have been evaluated using the **same preference data**. We will explicitly state this point in the revised manuscript.
>
> > Additionally, there is no ablation study comparing rule-based vs. LLM-generated preferences for O-MAPL
>
> We appreciate the reviewer's suggestion. To clarify, we already have Table 1 and several figures in the appendix which show a clear comparative analysis between **rule-based and LLM-based preference data** across all methods considered.
>
> > Implementation details lacked critical specifics on preference dataset generation (e.g., LLM hyperparameters) and omitted pseudocode, hindering reproducibility.
>
> We thank the reviewer for this feedback. To clarify, we have already included several details in the appendix on our data generation methods (rule-based and LLM-based), including hyperparameters and specific prompts used to obtain LLM-generated preferences. We believe this information is sufficient for reproducibility.
>
> Regarding the pseudocode, we agree that our algorithm description is currently quite brief. We will include a more detailed and structured presentation of the algorithm in the updated paper to improve clarity and reproducibility.
>
> > Additional experiments provided limited ablation studies (e.g., linear mixing tested only on SMAC, not MAMuJoCo) and did not address computational costs or robustness to preference noise
>
> We thank the reviewer for raising these important points. To clarify, our experiments did apply linear mixing and our proposed value decomposition methods to **both SMAC and MAMuJoCo tasks**.
> Regarding computational costs, we have provided some details in section B.2. If the reviewer could specify which particular aspects of preference noise we have overlooked, we would be happy to provide further clarification and analysis.
>
> > The work focuses exclusively on cooperative settings, leaving mixed cooperative-competitive environments unexplored. …
>
> We thank the reviewer for highlighting this important consideration. Our current work focuses specifically on cooperative settings. As noted in our conclusion, addressing mixed cooperative-competitive scenarios would indeed necessitate significantly different methodologies, which lie beyond the current scope. We agree that exploring these mixed scenarios is a valuable direction for future research.
>
> > The experiments do not include comparisons with non-linear mixing variants. …
>
> We thank the reviewer for this insightful comment. We focused our experiments on the linear mixing setting. This decision was based on previous studies indicating that two-layer mixing networks perform significantly worse than one-layer linear mixing networks especially in offline settings with limited data [1]
>
> [1] Bui et al. ComaDICE: Offline cooperative multi-agent reinforcement learning with stationary distribution shift regularization. In ICLR 2025.
>
> To address your question, we have conducted additional comparative experiments between linear and nonlinear mixing structures. The results, shown in **Table 3** in the **PDF**, clearly demonstrate that the linear mixing structure outperforms the two-layer structure.
>
> > The paper does not analyze the quality of LLM-generated labels.... How sensitive is O-MAPL to potential inaccuracies or biases in LLM-generated preferences?
>
> We thank the reviewer for the insightful comment. To address this concern, we conducted additional experiments comparing the performance of our method using two versions of ChatGPT: **4o and 4o-mini** (the latter being considered weaker in terms of long-term reasoning capabilities). The comparison results, reported in **Table 4** in the **PDF**.
>
> ---
>
> *We hope that the above responses satisfactorily address your questions and concerns. If you have further questions, we are happy to provide additional clarification.*

---

### Official Review · Reviewer_TgVH · 2025-03-13

**Overall Recommendation:** 4

**Summary:**

This paper proposes a multi-agent offline reinforcement learning algorithm named O-MAPL, addressing the problem of directly training multi-agent policies using human preference data under offline data conditions.
The authors highlight limitations of traditional two-stage methods (first learning a reward model, then learning a policy), including:
1. The requirement for large preference datasets covering state and action spaces.
2. Performance degradation due to misalignment between the two stages.

To address these issues, they propose an end-to-end framework that directly learns policies from preference data in an offline setting. Specifically, leveraging the correspondence between soft Q-functions and reward functions under the Maximum Entropy Reinforcement Learning (MaxEnt RL) framework, they introduce a method to train Q-functions (i.e., agent policies) directly using preference data.

Key technical contributions include:
1. A mixing network for value factorization to enable multi-agent coordination.
2. A linear value decomposition network designed to align with preference learning objectives and the IGM (Individual-Global-Maximization) principle, ensuring training stability and global-local consistency.

Experimental results demonstrate that O-MAPL outperforms existing methods across benchmark environments such as SMAC (StarCraft Multi-Agent Challenge) and MaMuJoCo (Multi-Agent MuJoCo).

**Claims And Evidence:**

The effectiveness of the proposed O-MAPL algorithm is sufficiently supported by experiments.

Validity of the end-to-end learning framework: Theoretical analysis and experimental results (on SMAC and MaMuJoCo benchmarks) demonstrate that learning policies directly from preference data outperforms traditional two-stage methods (baselines include single-agent methods extended to multi-agent settings, e.g., IPPL, SL-MARL, and methods similar to this work but lacking the proposed value decomposition architecture, e.g., IPS-VDN).

Importance of value decomposition: Theoretical analysis of the convexity of single-layer mixing networks and global-local consistency.

**Essential References Not Discussed:**

The related work section adequately covers offline RL and preference learning.
This work builds upon multi-agent offline RL and integrates preference data for policy training.

**Experimental Designs Or Analyses:**

1. Experiments are conducted on widely used SMAC and MaMuJoCo environments, ensuring representativeness.
2. Training with rule-based and LLM-generated preference data covers diverse feedback quality.
3. Results are statistically significant, with mean values and standard deviations provided.

**Methods And Evaluation Criteria:**

The proposed approach of directly learning multi-agent policies from preference data in an offline setting is meaningful for multi-agent preference learning. In multi-agent scenarios, obtaining accurate reward signals is challenging, and preference learning offers a promising solution. However, preference learning in MARL remains underexplored and faces issues such as training efficiency and human involvement costs. Existing two-stage methods (first learning a reward function, then training policies) require extensive preference data and suffer from misalignment between stages. Thus, directly training policies from preference data is a significant contribution to multi-agent preference learning. The use of SMAC and MaMuJoCo for evaluation is appropriate, as these environments are widely adopted for benchmarking multi-agent reinforcement learning methods. Training with both rule-based and LLM-generated preference data is reasonable, as these methods generate preference data of varying quality, enabling comprehensive algorithm evaluation.

**Other Comments Or Suggestions:**

Suggest adding implementation details in the appendix to facilitate reproducibility.

**Other Strengths And Weaknesses:**

Strengths:
1. Practical relevance: The offline learning paradigm avoids costly trial-and-error interactions.
2. Novelty: A new paradigm for multi-agent preference learning that mitigates defects of two-stage methods.
3. Rigorous theoretical analysis and experiments.

Weaknesses:
1. No evaluation under limited preference data, a common real-world constraint.
2. Insufficient demonstration of the value decomposition module’s specific contributions.

**Questions For Authors:**

1. How does the method perform when preference feedback is limited?
2. Can the approach be extended to mixed cooperative-competitive scenarios?

**Relation To Broader Scientific Literature:**

The proposed method is closely related to existing literature:
1. It extends the MaxEnt RL framework to multi-agent offline preference learning, enabling end-to-end training.
2. It aligns with preference learning research by leveraging preference data for multi-agent policy training.

**Theoretical Claims:**

The theoretical analysis in Section 4 includes:
1. Theorem 4.1 (Convexity Analysis): The loss function under a single-layer linear mixing network is convex.
2. Proposition 4.2 (Non-convexity in Nonlinear Networks): Two-layer mixing networks disrupt convexity.
3. Proposition 4.3 (Global-Local Consistency): Local policy optimization aligns with global objectives.
These theoretical analyses are clear and free of obvious errors.

---

> ### Author Rebuttal · Authors · 2025-03-31
>
> We sincerely thank the reviewer for the positive evaluation and insightful feedback. Below, we provide detailed responses to your comments. To address your questions, we have conducted additional experiments, which are detailed in the **PDF** available via this anonymous link: https://1drv.ms/b/s!AgChHLa7t5Bza5QJpMfi7YJX6PI
>
> ---
>
> > How does the method perform when preference feedback is limited?
>
> We thank the reviewer for the insightful comment. To address your question, we conducted additional experiments in which we systematically reduced the amount of preference data to 75%, 50%, and 25% of what was used in our main experiments. The comparison results — reported in Tables 1 and 2 in the **PDF** — demonstrate how our method and several important baselines perform under reduced preference feedback. As expected, the overall performance of all methods degraded with less data; however, O-MAPL consistently outperformed the other baselines across all settings, showing robustness to limited preference feedback.
>
> > Insufficient demonstration of the value decomposition module’s specific contributions.
>
> We thank the reviewer for the comment. To clarify, we have included several variants of our method to evaluate the specific contributions of the value decomposition module. Specifically, IPL-VDN is a variant where we fix the weights in the value decomposition module and do not learn them, while IIPL represents a variant in which each local value function is learned independently per agent, without being aggregated through our mixing architecture.
>
> In addition, we have conducted additional experiments comparing the performance of *1-layer* and *2-layer* mixing networks in our value decomposition approach. The comparison results are reported in Table 3 of the **PDF**.
>
> > Can the approach be extended to mixed cooperative-competitive scenarios?
>
> We thank the reviewer for the question. While the value decomposition approach has potential applicability to mixed cooperative-competitive scenarios, it would require substantial adaptations to handle such complexities effectively. We consider this extension beyond the current scope but agree it is a valuable direction for future exploration.
>
> ---
>
> *We hope that the above responses satisfactorily address your questions and concerns. If you have further questions, we are happy to provide additional clarification.*

---

### Official Review · Reviewer_xng9 · 2025-03-24

**Overall Recommendation:** 2

**Summary:**

The paper introduces O-MAPL, a novel framework for multi-agent reinforcement learning that leverages human preference data to train cooperative agents without explicit reward modeling. Traditional MARL methods often require separate stages for reward learning and policy optimization, leading to instability and inefficiency. O-MAPL addresses this by directly learning the soft Q-function from pairwise trajectory preferences, using a centralized training with decentralized execution (CTDE) paradigm. The approach incorporates a value decomposition strategy to ensure global-local consistency and convexity in the learning objective, enabling stable and efficient policy training.

The authors conduct extensive experiments on two benchmarks. Results show that O-MAPL consistently outperforms existing methods across various tasks, demonstrating its effectiveness in complex multi-agent environments. The paper also highlights the potential of using LLMs for generating rich and cost-effective preference data, which significantly improves policy learning.

Key contributions include:
1. An end-to-end preference-based learning framework for MARL that avoids explicit reward modeling.
2. A value factorization method that ensures global-local consistency and convexity in the learning objective.
3. Extensive empirical validation showing superior performance over existing methods.

**Claims And Evidence:**

Yes

**Essential References Not Discussed:**

No

**Experimental Designs Or Analyses:**

Yes

**Methods And Evaluation Criteria:**

Yes

**Other Comments Or Suggestions:**

I think how to guarantee the stability and convergence of the implicit-reward method can be critical to the proposed method.

**Other Strengths And Weaknesses:**

I have one concern about the correctness of the derivation of the implicit-reward method. In section 4.2, the authors proposed to design a training objective function $L(q,v,\omega)$ to avoid explicit reward modeling. However, this training objective relies on the inverse
soft Bellman-operator. In $L(q,v,\omega)$, I did not see how to guarantee the inverse soft Bellman-operator which requires the soft Q function to be a fixed point of a soft Bellman-operator. Therefore, in the training pipeline, if the updated Q function can not be guaranteed to be a fixed point of a soft Bellman-operator, then the algorithm can not provide a convergence guarantee to an optimal policy. In practice, this issue usually lead to instability or sub-optimal solutions of the implicit-reward methods [1, 2].

[1] Garg, Divyansh, et al. "Iq-learn: Inverse soft-q learning for imitation." Advances in Neural Information Processing Systems 34 (2021): 4028-4039.

[2] Rafailov, Rafael, et al. "Direct preference optimization: Your language model is secretly a reward model." Advances in Neural Information Processing Systems 36 (2023): 53728-53741.

**Questions For Authors:**

it will be helpful if the authors show that the Q function updated in their designed training pipeline can be a fixed point to one soft Bellman operator.

**Relation To Broader Scientific Literature:**

This paper considers a natural extension from existing implicit-reward offline RL method to MARL setting.

**Theoretical Claims:**

Yes

---

> ### Author Rebuttal · Authors · 2025-03-31
>
> We thank the reviewer for carefully reading our paper and providing valuable feedback. Below, we address your concern in detail.
> > I have one concern about the correctness of the derivation of the implicit-reward method. In section 4.2, the authors proposed to design a training objective function to avoid explicit reward modeling. However, this training objective relies on the inverse soft Bellman-operator. In L(), I did not see how to guarantee the inverse soft Bellman-operator which requires the soft Q function to be a fixed point of a soft Bellman-operator. Therefore, in the training pipeline, if the updated Q function can not be guaranteed to be a fixed point of a soft Bellman-operator, then the algorithm can not provide a convergence guarantee to an optimal policy. In practice, this issue usually lead to instability or sub-optimal solutions of the implicit-reward methods [1, 2].
>
> We thank the reviewer for the comment. To clarify, we can formally prove that the trained $Q_{\text{tot}}$ resulting from our preference-based training objective is guaranteed to be a fixed-point solution of the soft Bellman equation. The proof closely follows the structure of prior work such as IQ-Learn, and we provide a version tailored to our formulation and notation in the following.
>
> **PROOF**:
> With the definition of $\mathcal{T}^* Q_{\text{tot}}$ as
>
> $(\mathcal{T}^* Q_{\text{tot}})(s, a)$ = $Q_{\text{tot}}(s, a)$ - $\gamma$ $E_{s'}$ $[V_{\text{tot}}(s')]$
>
> we can rewrite the inverse Bellman equation as follows:
>
> $Q_{\text{tot}}(s, a)$ = $(\mathcal{T}^* Q_{\text{tot}})(s, a)$ + $\gamma$ $E_{s'}$ $\left[ V_{\text{tot}}(s') \right]$,
>
> which implies:
>
> $Q_{\text{tot}}(s, a)$ = $(B_r^*$ $Q_{\text{tot}})(s, a)$,
>
> where the reward function is defined as
>
> $r(s, a)$ = $(\mathcal{T}^* Q_{\text{tot}})(s, a)$, and $(B_r^*$ $Q_{\text{tot}})(s, a)$ is the soft Bellman backup operator defined in our paper. This result shows that $Q_{\text{tot}}$ satisfies the soft Bellman equation.
>
> Therefore,  if we train the preference-based objective using $(\mathcal{T}^* Q_{\text{tot}})(s, a)$ — i.e., via the inverse soft Bellman equation — then the resulting $Q_{\text{tot}}$ is guaranteed to be a **fixed-point solution** to the soft Bellman equation with reward $r(s, a)$ = $(\mathcal{T}^*$ $Q_{\text{tot}})(s, a)$.
>
> **EndProof**
>
> We will incorporate the above discussion into the revised paper to clarify the fixed-point convergence of our method.
>
> ---
>
> *We hope that the above responses satisfactorily address your questions and concerns. If you have further questions, we are happy to provide additional clarification.*

---

### Decision · Program_Chairs · 2025-05-01

**Decision:**

Accept (poster)

**Comment:**

he paper introduces O-MAPLE, an end-to-end framework for offline multi-agent reinforcement learning (MARL) from preferences, avoiding explicit reward modeling by leveraging soft Q-function estimation via inverse Bellman operators. The proposed framework integrates value decomposition with convex mixing networks and uses centralized training with decentralized execution (CTDE), addressing scalability and stability challenges in preference-based MARL. The paper includes theoretical analyses, experimental validation across standard MARL benchmarks (SMAC, MaMuJoCo), and studies the utility of LLM-generated preferences.

There is some disagreement regarding the interpretation of regularized value functions and whether the Q-function must be optimal for the inverse Bellman operator to be meaningful. The authors are encouraged to clarify the theoretical assumptions and limitations surrounding the use of the inverse soft Bellman operator when the Q-function is not yet optimal. While the derivation appears sound, a discussion of convergence conditions and practical implications would benefit readers.